# Quantifying complexity in DNA structures with high resolution Atomic Force Microscopy

Elizabeth P. Holmes[1,10], Max C. Gamill [1,10], James I. Provan[2,8,10], Laura Wiggins[1], Renáta Rusková [3], Sylvia Whittle [1], Thomas E. Catley [1], Kavit H. S. Main[4], Neil Shephard [5], Helen. E. Bryant [6], Neville S. Gilhooly[7,9], Agnieszka Gambus [7], Dušan Račko [3], Sean D. Colloms [2] ✉ & Alice L. B. Pyne [1] ✉

DNA topology is essential for regulating cellular processes and maintaining genome stability, yet it is challenging to quantify due to the size and complexity of topologically constrained DNA molecules. By combining high-resolution Atomic Force Microscopy (AFM) with a new high-throughput automated pipeline, we can quantify the length, conformation, and topology of individual complex DNA molecules with sub-molecular resolution. Our pipeline uses deep-learning methods to trace the backbone of individual DNA molecules and identify crossing points, efficiently determining which segment passes over which. We use this pipeline to determine the structure of stalled replication intermediates from *Xenopus* egg extracts, including theta structures and late replication products, and the topology of plasmids, knots and catenanes from the *E. coli* Xer recombination system. We use coarse-grained simulations to quantify the effect of surface immobilisation on twist-writhe partitioning. Our pipeline opens avenues for understanding how fundamental biological processes are regulated by DNA topology.

The complex topological landscape of DNA is essential to cellular function. For example, while most DNA is negatively supercoiled, positive supercoiling at transcription start sites has been shown to influence mRNA synthesis[1]. DNA can also become tangled, either with itself or with other DNA molecules. This entanglement can impede replication and transcription, increase DNA damage and mutation rates, prevent chromatin assembly, and may even influence cellular differentiation[2–7]. The misregulation of DNA topology is an event which can trigger cell death at cytokinesis, thus ensuring that genome

integrity is maintained[8,9]. Defects in the regulation of DNA topology can also lead to disease such as cancer and neurodegeneration[10–12]. To fully understand the role of DNA topology, including its regulatory processes and pathologies, we must develop new methods to determine how the mechanical and geometric properties of DNA affect its interactions with other biomolecules. DNA topology encompasses both its superhelical properties (over- or under-winding of the DNA helix) and its entanglement. Knots are self-entangled individual DNA circles, while catenanes consist of two or more interlinked circles of

[1]School of Chemical, Materials and Biological Engineering, University of Sheffield, Sheffield, UK. [2]School of Molecular Biosciences, University of Glasgow, Glasgow, UK. [3]Polymer Institute of the Slovak Academy of Sciences, Bratislava, Slovakia. [4]London Centre for Nanotechnology, University College London, London, UK. [5]School of Computer Science, University of Sheffield, Sheffield, UK. [6]School of Medicine and Population Health, University of Sheffield, Sheffield, UK. [7]Department of Cancer and Genomic Sciences, University of Birmingham, Birmingham, UK. [8]Present address: Institute for Integrative Biology of the Cell (I2BC), Université Paris-Saclay, Gif-sur-Yvette, France. [9]Present address: Oxford Nanopore Technologies plc, Gosling Building, Edmund Halley Road, Oxford Science Park, Oxford OX4 4DQ, UK. [10]These authors contributed equally: Elizabeth P. Holmes, Max C. Gamill, James I. Provan. ✉e-mail: sean.colloms@glasgow.ac.uk; a.l.pyne@sheffield.ac.uk

DNA[13] (Supplementary Fig. 1a). These structures can be inferred at the ensemble or single molecule levels from their biophysical properties by, e.g. electrophoretic techniques[13–27], DNA looping assays[28], optical and magnetic tweezer measurements[29], and nanopore detectors[30–33]. Of these, gel electrophoresis is by far the most accessible and well-defined method[34–38]. In the absence of supercoiling, topological species migrate according to topological complexity or average crossing number in 3D space. Thus, if two distinct topological species of the same size have a similar average crossing number (e.g. a 4-node catenane and a 4-node knot) they will migrate at a similar position on the gel. In contrast, microscopy techniques, such as Atomic Force Microscopy (AFM)[26,39–45] or electron microscopy (EM)[13,17,21–23,46–53], provide full geometrical descriptions of the structure of single DNA molecules, allowing precise topological determination in addition to information on local curvature[54,55], and helical structure.

Microscopy-based determination of entangled DNA must faithfully trace the path of the DNA molecules and then discriminate the "crossing order" of intersecting segments i.e., which segment passes over which at each crossing. Typically, the technique of choice has been transmission EM of rotary-shadowed specimens, where a DNA sample is coated with RecA, shadowed, and then imaged under vacuum[13,49,51–53,56]. This technique helped elucidate the topological reaction mechanisms of various site-specific DNA recombinases and topoisomerases[17,22,23,50,57,58], but is technically demanding, and can suffer from problems with inconsistent rotary-shadowing and incomplete RecA polymerisation on double stranded DNA, with some crossing geometries remaining ambiguous.

AFM imaging has become a powerful tool to probe the structure and interactions of DNA[39,42–44,59–63]. AFM provides nanometre resolution imaging on single molecules in aqueous conditions with real-time imaging capabilities, and minimal sample preparation. Recent studies have made use of AFM to mechanistically explore DNA-condensin interactions[64], the binding kinetics of a transposition complex[63], and the effect of supercoiling density on DNA minicircle structure[43]. However, no studies to date have exploited the three-dimensional capabilities of AFM to determine the crossing order where one DNA duplex crosses another in topologically complex species, or have done so in an automated manner that allows this to be quantified at scale.

Here we present an automated pipeline for the high-throughput tracing of single, untreated DNA molecules from high resolution AFM images, captured in aqueous conditions. Our analysis tool traces each DNA molecule, identifies each point where it crosses itself and defines which segment passes over which, i.e. the crossing order, by analysing the differential height profiles of under- and over-passing DNA. This enables topological classification through the Python package Topoly[65].

To determine the effectiveness of our pipeline on complex DNA structures, we quantify the composition of DNA replication intermediates from *Xenopus laevis* egg extracts stalled with two orthogonal model replication fork impediments, the Lac repressor and Tus protein[66–69]. We use high-resolution AFM imaging to visualise the entire structure and use our automated pipeline to calculate the contour length of the entire structure. Beyond this, we automatically identify replication forks and quantify the length of the unreplicated DNA between them. Furthermore, we identify a number of additional structural features, including stalled forks, and quantify their length and frequency, giving additional information beyond global structural composition.

To test the fidelity of our pipeline, we utilise the *E. coli* Xer recombination system (Supplementary Figs. 1–6) in vitro to generate a suite of plasmid-sized predictable topological products including homogeneous right-handed 4-node catenanes, and knots of increasing node-number[18,70–74]. Using these we construct a thorough representation and classification of the behaviour and topology of DNA by AFM. In addition, we quantify a recurrent depositional effect observed during our examination of 4-node catenated DNA; where the clustering of crossings obscures the overall conformation of the molecule. We further explain this effect by the comparison of our AFM observations to predictions of coarse-grained molecular dynamics simulations. By providing an objective and efficient means to explore the conformation and topology of varied DNA structures, we can interrogate fundamental processes involving DNA-protein transactions, with potential impact on a range of cellular processes.

## Results

### High-resolution AFM enables accurate tracing and contour length measurement via crossing order determination

High-resolution AFM imaging enables visualisation of the double helix of DNA on individual molecules in aqueous conditions, without manipulation beyond the process of deposition onto a flat mica substrate. The height information that AFM provides should enable us to determine which DNA duplex passes over and under at each crossing (the "crossing order"), and therefore explicitly determine the topology of individual molecules. We use the defined product of the *E. coli* Xer recombination system between two directly repeated *psi* sites on plasmid p4CAT (1651 bp); a right-handed 4-node catenane, consisting of one larger (1253 bp) and one smaller (398 bp) DNA circle (Supplementary Fig. 1c). Visual inspection of a high-resolution AFM image shows that the product contains the expected large and small circles, interlinked by 4 well-separated crossings (#1-4), with the addition of one 'trivial' self-crossing (#5) within the large circle that does not contribute to the overall topology (Fig. 1a). If each crossing is assigned as over- or under-passing by eye, the catenane appears to have the predicted topology (Fig. 1b).

To define the path each DNA duplex takes through the product and the overall DNA topology, we must determine the crossing order for every intersecting duplex. The over-passing segment will form a "humpback bridge" conformation, as the under-passing segment passes underneath. We use the height information traced through each crossing to determine the crossing order as the over-passing profile should form a wider peak than the under-passing (Fig. 1c). To test this, the height is traced manually along each duplex through the crossing, using the AFM processing software Gwyddion[75], with the wider peak assigned as the over-passing duplex, and the narrower the under-passing (Fig. 1d). When assigned in this manner, the crossings alternate between over and under as each circle is traced (the small circle overlies the large circle at crossings 1 and 3 but underlies it at crossings 2 and 4). As we cannot reliably determine DNA sequence direction from the AFM images, we cannot absolutely determine the sign notation at each crossing. Independent of sequence orientation, there are two possible right-handed topologies for this 4-node catenane (Supplementary Fig. 1a-iii). If we arbitrarily assign antiparallel orientations to the two circles, the four right-handed crossings will have a negative crossing sign (Supplementary Fig. 1a-iii). This corresponds to the expected right-handed anti-parallel 4-node topology of the catenane (Fig. 1b), which has previously been determined by biochemical methods[71], confirming our visual assignment of the crossing order. In this example, the misassignment of a single crossing order would lead to incorrect classification of the topology as a 2-noded catenane, while misassignment of two crossings would lead to classification as two unlinked circles (Supplementary Fig. 3).

However, manual assignment of topology is a prolonged process and subject to observer bias. Additional challenges include tracing the entire molecular backbone and incorporating the measurements of individual crossings into the wider molecular architecture to extract topological characteristics such as writhe. To remove this bottleneck, we develop an automated pipeline to trace and classify individual molecules and determine their length and topology. We image a series of topologically complex molecules, including nicked and negatively supercoiled plasmids and catenated and nicked knotted DNA (Fig. 1e).

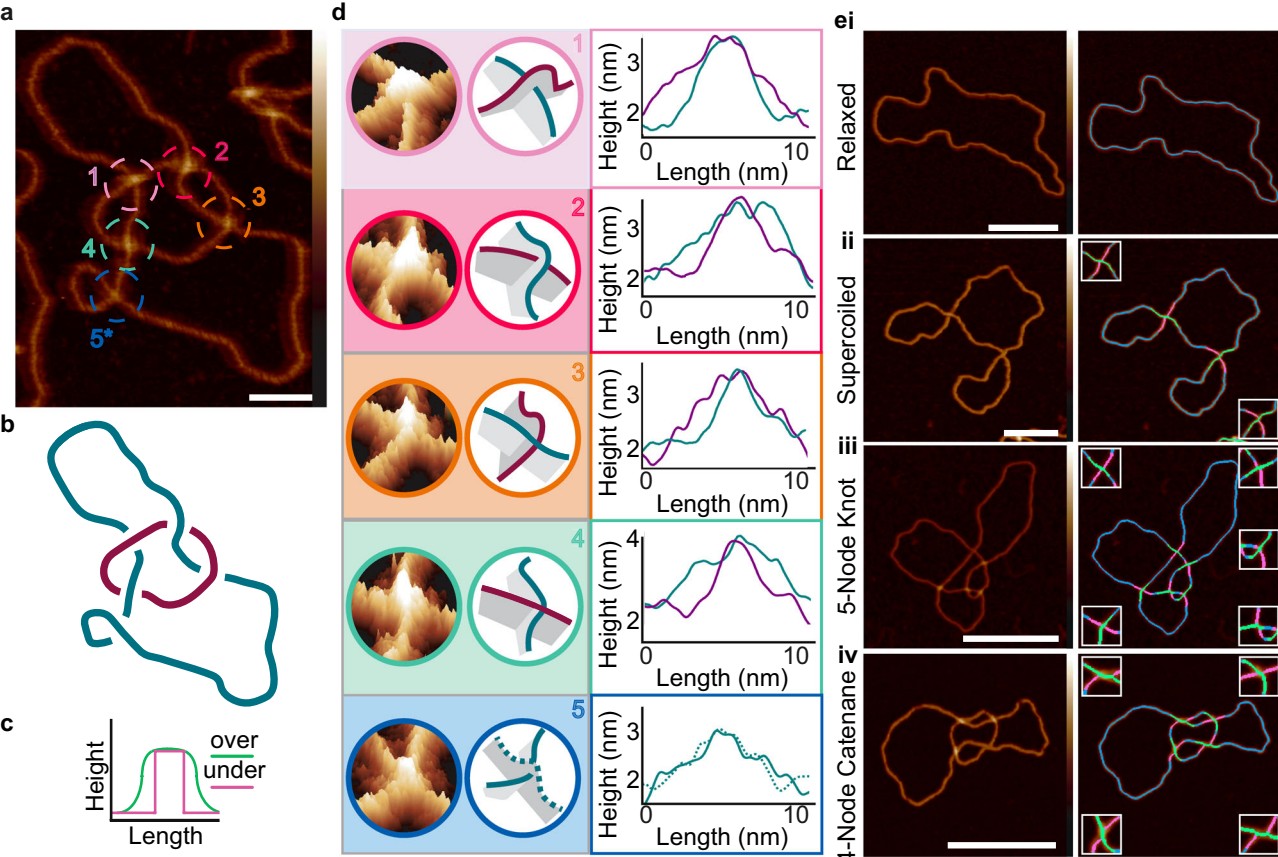

**Fig. 1 | Determining the topology of individual DNA molecules using AFM.**
**a** 4-node DNA catenane visualised using AFM with helical structure visible (of
$N = 1010$ total DNA plasmid, knot and catenane molecules imaged). Scale bar:
20 nm. Colour bar: −2 to 6 nm. **b** Schematic of the right-handed 4-node catenane in
**a** with crossing orders determined by eye. **c** Schematic of the expected crossing
height profiles of over-(lime green) and under-passing (hot pink) DNA duplexes at a
crossing. **d** tilted 3D height profiles of each catenane crossing, schematics of each
crossing in 3D, and manual line profiles of each crossing (numbered as in **a**). The
large circle is coloured blue and small purple. The crossing order of each crossing is

determined visually and agrees with the known topology of the catenane.
**e** Automated tracing and topological determination of DNA structures from AFM
images, for: **i**, relaxed ($N = 67$) and; **ii**, supercoiled plasmid (2260 bp) ($N = 78$); **iii**,
5-node twist knot produced from pDIR (2260 bp) ($N = 108$) and; **iv**, catenated
(1253 bp, 398 bp) plasmids produced from p4CAT ($N = 604$). Automated traces
(blue) show under-passing (hot pink) and over-passing (lime) segments at cross-
ings. Scale bar: 50 nm. Height scale: −2 to 6 nm. Source data is provided as a Source
Data file.

To enable accurate tracing analysis and crossing detection, the mole-
cules should be in as open a conformation as possible. A buffer solu-
tion containing magnesium chloride in place of nickel chloride[60,76,77] is
used to adhere the DNA to the mica surface. This buffer immobilised
the DNA in a more open conformation (Supplementary Fig. 7a–d),
lowering the chance of trivial self-crossings. Given the level of super-
coiling of the unknot plasmid ($\sigma = -0.06$), the expected writhe should
be on the order of 10 self crossings[78]. However, due to the immobili-
sation in $MgCl_2$, we only see $1.7 \pm 1.4$ visible self-crossings within the
supercoiled unknot sample. By comparison, for the sample with $NiCl_2$
immobilisation, $4.4 \pm 2.3$ self-crossings are visible (Supplementary
Fig. 7f). We measure the bounding area of the molecules and observe
that the nicked molecules immobilised in $MgCl_2$ adopt the most open
conformations, and the supercoiled molecules in $NiCl_2$ the most
closed conformations (Supplementary Fig. 7e).

Our accurate tracing of the DNA molecules allows us to determine
the mean contour length and standard deviation for each sample of
unknotted, knotted and catenated structures. For the simplest struc-
ture, unknotted plasmids, the contour length is within 1 % of the
expected length ($761 \pm 11$ nm vs 768 nm expected) (Supplementary
Table 1). However, for more complex molecules, multiple DNA cross-
ings cluster into a singular crossing reducing the number of correctly
identified molecules obtained after data cleaning steps. This is evi-
denced by a larger proportion of retained molecules in the 3-node

knots (56%), which have a lower number of crossings and hence a lower
propensity to cluster, than the 5-twist knots (7%) (Supplementary
Table 1). The clustering also affects catenated molecules, where 80 of
145 catenated structures can be separated into their individual mole-
cules. However, for those that are separable, contour lengths are
determined to within 5% of their expected length (Supplementary
Fig. 8). Beyond contour length, we determine the number of crossings
in each molecule and therefore infer whether molecules of the same
topology are likely to be supercoiled (Supplementary Figs. 7f, 9).

### An automated pipeline for molecular tracing and explicit determination of DNA topology

To further enhance the scope of our methodology, we wanted to
automate the topological determination of complex molecules as well
as measurements of their contour length. There are two major chal-
lenges in automating the classification of DNA topologies from AFM
images: accurate path tracing, which can discriminate between adja-
cent or crossing DNA duplexes, and the determination of the crossing
order at each crossing. Therefore, we developed multiple new meth-
ods in this work to overcome these challenges and implemented them
into the open-source software TopoStats[79] to maximise their
accessibility.

Self-crossing or close-passing molecules cannot be segmented
reliably using traditional methods such as binary thresholding (Fig. 2a;

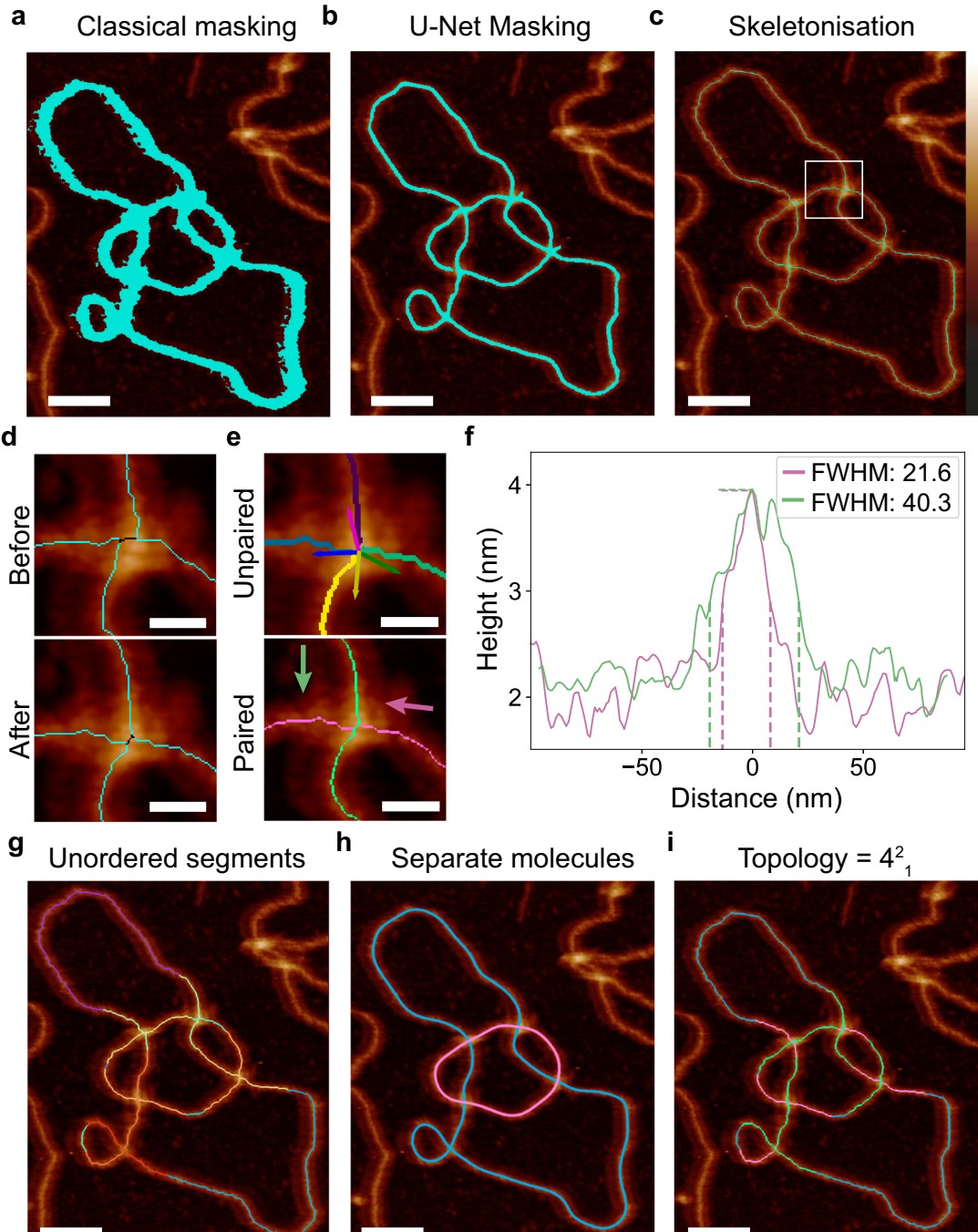

**Fig. 2 | Automated tracing and topological determination for complex DNA molecules from AFM images.** Pre-processed AFM images are masked to identify molecules (of $N = 1010$ total DNA plasmid, knot and catenane molecules imaged) using (**a**) classical image processing and **b** deep learning methods (U-Net masking). **c** An enhanced skeletonisation method, using height-biasing, reduces binary masks to single pixel traces along the centre of the molecule, and **d** locates the central point or node of the crossings (black), minimising misalignment of emanating branches. **e** Emanating branches (purple, pink, blue and dark green) are paired based on their propagating vectors and a height trace spanning the crossing segment (light green and pink) is obtained. Arrows show the propagation direction of the segments. **f** The full-width half-maximum (FWHM) of each crossing is used to determine the crossing order of the segments, with the maximal FWHM designated as over-passing. The over-passing segment can be seen in green while the under-passing segment can be seen in pink. Dotted lines illustrate their FWHM measurements. **g**, **h** Branch pairing at each node enables separation of entangled molecules, shown here in magenta and cyan. **i** Under- and over-passing crossing classifications allow for topological classification which is output in Rolfsen knot notation format[82], where $4^2_1$ correctly denotes the 4-node DNA catenane formed of two circular molecules that cross one another a minimum of four times. Scale bar: **a**–**c**: 50 nm, **d**, **e**: 10 nm, **g**–**i**: 50 nm. Height scale (**c**): −2 to 6 nm. $N = 1$ repeat for this image. Source data are provided as a Source Data file.

Supplementary Figs. 10 and 11). Therefore, we train a deep-learning U-Net[80] model to perform image segmentation for DNA molecules, producing clearer, more reliable segmentations (Fig. 2b). We then skeletonise the segmented image to produce single pixel traces which

follow the molecular backbone (Fig. 2c). To faithfully recapitulate the path of DNA through molecular crossings, we enhance the Zhang and Shuen[81] skeletonisation algorithm to take advantage of the height data present in AFM images, to 'bias' the skeletonisation onto the centre of

the molecule, even at crossings (Fig. 2d). This is a substantial improvement over the manual tracing described above, which took straight-line profiles through crossings, reducing the accuracy of the crossing order determination. We perform convolution to detect the central point or "node" (Fig. 2d - black) of the crossing, and pair the emanating branches as described in the methods to obtain continuous molecular traces (Fig. 2e), even for complex DNA molecules with several intersections.

To determine the crossing order, we classify the height profile of the DNA duplex with the greatest full-width half-maximum (FWHM) through the crossing as over-passing (Fig. 2f - green). The FWHM metric is chosen over a simpler implementation e.g., average height, or area under the curve, because the resulting classification from these metrics has greater influence from trace artefacts such as close nodes or nicks, causing inaccuracies. Secondly, FWHM does not require Gaussian curve fitting, which is challenging to auto-fit across a dataset, particularly when trying to differentiate from the noisy background trace. We quantify the average crossing order reliability for each crossing using the ratio between the minimum and maximum FWHM values ($FWHM\_pair$) for $N$ paired branches (Eq. 1). Using a ratio of the true positive to false negative classifications for 83 hand-labelled crossings and the code-produced crossing orders, we suggest a crossing reliability threshold of 0.263 optimises the number of correct to incorrect classifications (Supplementary Fig. 12).

$$average\ crossing\ order\ reliability = \frac{1}{N}\Sigma_{FWHM_{pair}} 1 - \frac{\min(FWHM_{pair})}{\max(FWHM_{pair})}$$

(1)

The path of a single circular molecule (knotted or unknotted) is traced in its entirety, starting and ending at a single point on the path. However, further complexity is introduced when multiple molecules are entangled, such as in DNA catenanes. For those images, the constituent molecules are separated and then traced (Fig. 2g, h). Single or entwined molecule traces are then classified using Rolfsen's knot notation format by Topoly[65,82] (Fig. 2i), e.g. $4^2_1$ for a 4-node catenane. Although we produce a topological classification for each molecule, the general sample topology can also be identified using other measures from the software e.g., the topology with the largest minimum crossing order reliability (Supplementary Fig. 13), or the topology most represented in the distribution (Supplementary Fig. 14).

Building on our existing software TopoStats[43,79], this work demonstrates the first integrated pipeline to identify, trace, and determine the topology of complex DNA molecules, including knotted, catenated and supercoiled DNA substrates. The improvements we have made to the tracing pipeline enable us to trace complex self-intersecting molecules for the first time (Supplementary Fig. 11). To test the tracing capabilities of this new pipeline, we apply it to the complex structures of plasmid replication intermediates, which are akin to theta curves. The complexity in these structures includes replication forks, which appear as nodes with an odd number of branches, where a single piece of un-replicated DNA emanates from an intersection with two newly replicated DNA duplexes.

**Structural analysis of *Xenopus* replication products reveal intermediate structures dependent on the stalling complex**
We apply our automated pipeline to determine the structure and composition of plasmids replicated, partially or fully, in *Xenopus laevis* nucleoplasmic egg extract. The well-characterised Lac repressor and Tus-Ter complexes[66–69] are used to stall or impede replication fork progression, (Fig. 3a). This yields partially replicated intermediates (theta structures) or late replication intermediates (figure-of-8 molecules[83]), which can be observed using AFM (Fig. 3b, c respectively).

Replication in the *Xenopus* extract system proceeds in both directions from the origin of replication in the plasmid due to bi-directional replication. This creates two identical replication forks unwinding the DNA, producing two equal-sized newly replicated segments with an unreplicated portion of DNA between them[84]. The Lac repressor protein binds extremely tightly to operator sequences, and arrays of LacI have previously been used to site-specifically stall replication forks in egg extracts due to the protein blockage on the DNA[69,85]. To determine whether we can use our pipeline to determine the extent of DNA replication, we sample two timepoints (40 and 80 minutes) after replication was initiated. For both timepoints, we observe that most of the molecules can be identified as theta structures (Fig. 3bi, bii), with two replication forks (arrow heads pointing inwards highlighting the replication fork) and a length of unreplicated DNA (green) between them. The replication forks can be identified as crossings with 3 emanating branches. By isolating these odd-branched crossing regions, we can trace each individual DNA segment and determine the contour length, and thus the length of unreplicated DNA.

Using our pipeline, we can clearly identify three-way junctions and measure the length of DNA between them (green traces) (Fig. 3biii,-biv). The lengths of each segment of DNA between a 3-way junction are calculated, with two newly replicated regions calculated as very close in length, and the unreplicated DNA length identified as the shortest segment between two forks. To ensure the fidelity of this measurement, we carry out stalling via an alternative complex, Tus-Ter, which forms a polar fork arrest in *E. coli* to prevent over-replication and has been used in mammalian cells as a non-polar fork stalling system[86]. In our conditions, Tus-Ter presents a less severe block to replication fork progression. The replication intermediate structures at 40 minutes (T40) show little to no unreplicated DNA (Fig. 3ci) and the 80-minute time point shows most molecules as fully replicated plasmids (Fig. 3cii). The T40 samples were dimeric in length, $2240 \pm 77$ nm, and present as a figure-of-8 structure. Whereas the structures stalled at 80 minutes appeared to be the length of the original plasmid, suggesting the plasmid had fully undergone replication (Fig. 3ciii, civ, g).

Additionally, some structures appear to contain forks with ssDNA gaps (Fig. 3d), which can be identified by resolving the height difference between the DNA duplex and the ssDNA where it joins the main replication fork (Fig. 3dii). We also observe reversed replication forks, although they are a rare occurrence (observed in 7.7% of overall structures). Replication forks are defined as 3-way junctions, whereas reversed replication forks are defined as 4-way junctions where one emanating segment is short (<50 nm in length). A reversed replication fork can occur where a replication fork encounters an obstacle such as a DNA lesion, which enables replication machinery to reverse its course, creating a 4-way junction to avoid replicating through the lesion and therefore avoiding genotoxic stress[87]. Our pipeline detects these forks and measures the length of the incomplete branch, which is made possible due to the high resolution of our data (Fig. 3diii, div). To further ensure the validity of our pipeline, we compare the manual traces of the reversed fork length with the automated trace outputs, revealing a remarkably similar output (Supplementary Fig. 15f).

There is an obvious difference in the composition of the theta structures at 40 and 80 minutes with LacI (Fig. 3e), with a reduction in the length of unreplicated DNA from $313 \pm 156$ nm (L40) to $218 \pm 153$ nm (L80). This correlates to an increase in the overall contour length of the theta structures, including replicated and unreplicated DNA (Fig. 3f), from $2400 \pm 168$ nm after 40 minutes to $2500 \pm 219$ nm after 80 minutes. This is consistent with LacI inducing a robust slowdown of replication fork progression rather than a terminal stall. At long time points, the array is eventually read-through, and the replication of a plasmid completed (Fig. 3a +LacI 120 minutes). Manual analysis of the unreplicated contour length shows a similar distribution to the outputs from our pipeline with a

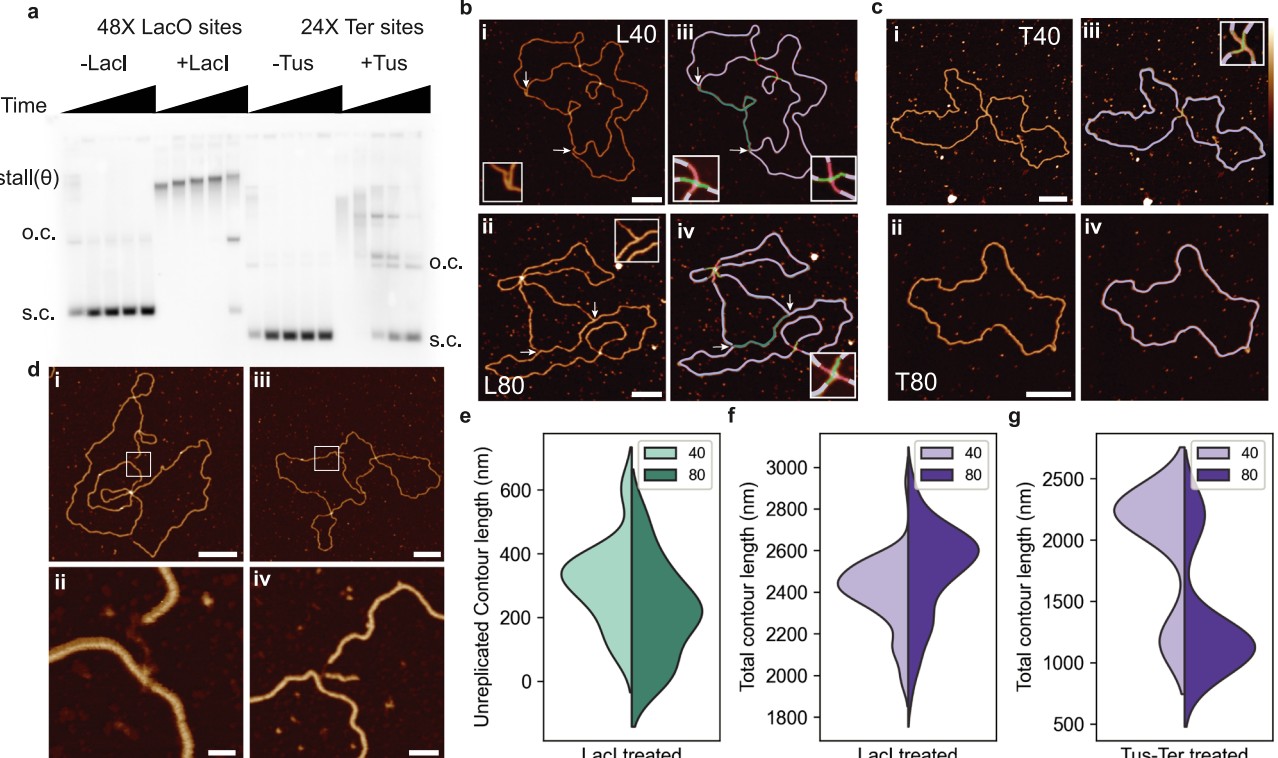

**Fig. 3 | Automated determination of DNA replication intermediates stalled using either the Lac Repressor protein or the Tus-Ter Complex, at 40 and 80 minute timepoints. a** Bulk DNA replication reaction products visualised on agarose gel. 4402 bp plasmid with 48 LacO sites (pJD97), or 3574 bp plasmid with 24 Tus sites (pUCattB-Ter24) are used as indicated. LacI or Tus proteins are added as indicated before initiation of the DNA replication reaction, which is then terminated after 10, 20, 40, 80 and 120 min. DNA samples are separated on agarose gel and nascent DNA is visualised by autoradiography. Replication products stalled at 40 and 80 minutes were imaged by AFM, with stalling performed using the (**b**) lac repressor protein (L40 N = 51, L80 N = 54) on a 4402 bp plasmid, and **c** Tus-Ter complex (T40 N = 20, T80 N = 18) on a 3574 bp plasmid. AFM micrographs of (**d**) stalled forks with **i** single-stranded DNA at the start of the fork, **ii**, including a high-resolution image of ssDNA gap. **iii**, a reversed replication fork, **iv**, including a high-resolution image of the reversed fork. Scale bars for **ii** and **iv** are 20 and 25 nm, respectively. Automated measurement of unreplicated DNA contour length, (**e**) 313 ± 156 nm, (L40, N = 12) and 218 ± 153 nm, (L80, N = 12), total contour length of lac repressor stalled DNA, (**f**) 2400 ± 168 nm, (L40, N = 51), 2500 ± 219 nm, (L80, N = 54) and Tus-Ter treated DNA (bimodal distribution) (**g**) 2240 ± 77 nm and 1170 ± 8 nm, (T40, N = 20), 1110 ± 90 nm and 2200 ± 83 nm (T80, N = 18). All singular values are mean ± SD, and bimodal values are calculated using a Gaussian mixture model with two components. Remaining scale bars: 100 nm. Height scale: −2 to 4 nm. Source data are provided as a Source Data file.

reduction in the contour length from 361 ± 56 nm to 196 ± 98 nm (Supplementary Fig. 15e).

For the Tus-Ter complex, a similar increase in replication is indicated after 80 minutes, however, this presents as an increase in the number of completely replicated plasmids, i.e., a reduction in overall contour length. The Tus-Ter stalled sample stalled at 80 minutes contains several fully replicated plasmids, measuring 1190 ± 90 nm in length, only 25 nm (2%) from the expected length of 1215 nm for a 3574 bp plasmid. This is approximately half the contour length measured at the T40 time point, further confirming the presence of the figure-of-8 molecules in that sample (Fig. 3g, Supplementary Fig. 15). This data indicates that stalling using the Tus-Ter complex results in late replication intermediates consistent with a termination stall, in comparison to LacI-induced fork blockades.

Having established that we can observe and quantify structural changes in replicating DNA plasmids, we now apply our pipeline to explicitly determine the topology of complex knotted DNAs formed by the synapsis of the Xer recombination accessory proteins PepA and ArgR.

### Automated topological identification of complex knotted DNA formed by Xer accessory proteins

The Xer accessory proteins PepA and ArgR form an interwrapped nucleoprotein synapse with a linear DNA substrate containing two *cer* recombination sites (Supplementary Fig. 2). With the addition of DNA ligase, the linear DNA is circularised and generates knotted species of

DNA, referred to as "*cer* ligation". The class of knots formed depend on the relative orientation of the *cer* sites on the substrate DNA; direct repeat sites form a mixture of twist-knot ligation products (e.g. $5_2^*$, $6_1^*$), while inverted repeat sites form specific chiral forms of torus knots (e.g. $3_1$ and $5_1$) and not their mirror images (Fig. 4a). The 5-node torus ($5_1$) and twist ($5_2$) knots run similarly on a gel (Fig. 4b), and the two chiral forms of each knot[88] (i.e. the same knot with positive or negative crossing signs) are indistinguishable by routine gel electrophoresis[72].

We obtained AFM images of an unknotted plasmid ($0_1$) and nicked 5-node knots gel-purified from the pINV and pDIR circularisation reactions, respectively which show distinct internal DNA crossings by eye (Fig. 4c–e). We trace each molecule using our pipeline to determine their length, crossing order, a reliability score for each crossing, and to classify the knot type (Fig. 4f–h). We determined that the supercoiled unknot plasmid contains two negative crossings as expected for negatively supercoiled, unknotted DNA (Fig. 4f, i). The product from the pDIR reaction contains 5 negative crossings, and is correctly identified as a 5-node twist knot ($5_2^*$) (Fig. 4g, j) while the product from pINV contains 5 positive crossings, and is correctly identified as a 5-node torus knot ($5_1$) (Fig. 4h, k).

Incorrect identification of the crossing order of a single crossing for these knotted products can lead to incorrect identification as either an unknot or a 3-node knot (Supplementary Figs. 5, 6, 16, and 17). At some crossings, one DNA duplex does not pass distinctly over the other, but instead the crossing has approximately the same height as a single DNA

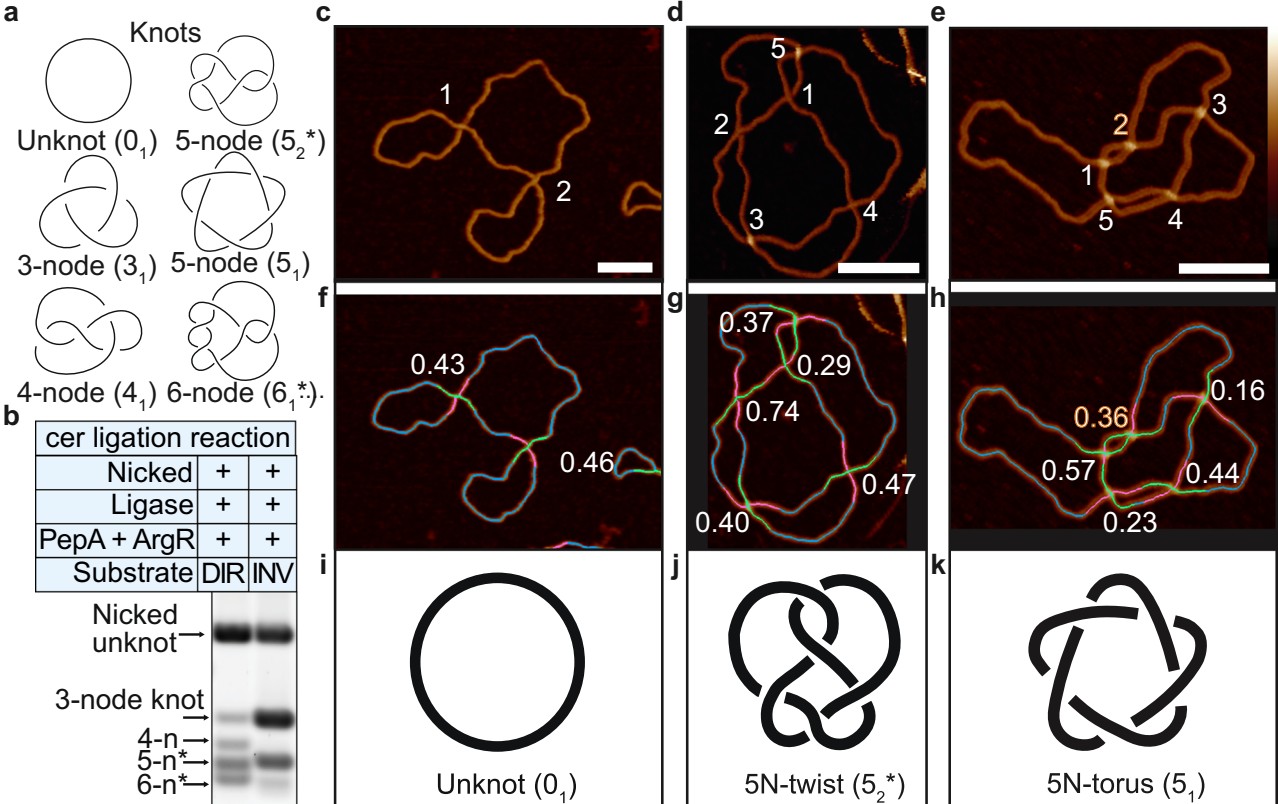

**Fig. 4 | Automated topological determination of twist and torus-type 5-node knots. a** Diagram of knots that can be created by the *cer* ligation reaction. See Supplementary Fig. 2 for proposed knotting mechanism (**b**), 1D gel electrophoresis of the knots shown in **a**. **c, d, e** AFM micrographs of unknot ($N = 145$), 5-node twist ($N = 108$), 5-node torus ($N = 89$) with crossings labelled 1–5 showing the order in which to trace the molecule starting from 1–5 with **f**–**h** corresponding automated traces ($N = 81$, $N = 18$, $N = 6$, respectively) showing under-passing (hot pink) and

over-passing (lime) segments at crossings. All crossing signs are negative in **f** and **g**, and positive in **h**. Schematics (**i**–**k**) of the unknot (**c**) and 5-node knot isolated from pDIR (**d**) and the 5-node knot isolated from pINV (**e**) (all 2260 bp). Automated traces of micrographs (**f**–**h**) are coloured to show the crossing order determined from the ratios of the FWHM. The numbers at each crossing are the crossing order reliabilities for each crossing. Scale bars: 50 nm. Colour Bar: −2 to 6 nm. Source data are provided as a Source Data file.

duplex, implying possible compression by the AFM tip or interdigitation of the DNA helices. This is marked by a ratio of FWHM close to 1 and a low crossing order reliability value for the crossing, increasing the difficulty of determining the correct topology for the molecule. We observe this by an increase in the correct topology when considering both the original topological classification and the classification when flipping the crossing order of the lowest reliable crossing (Supplementary Fig. 13).

Molecules with a higher number of expected crossings will therefore have a lower theoretical probability of obtaining the correct topological classification, which can be calculated using combinatorics. This results in a 56% probability of correct classification for 3-node knots, reducing to 37% for more topologically complex 5-node knots (Supplementary Fig. 18). We obtain an 82% crossing order accuracy from our hand-labelled crossing comparison (Supplementary Fig. 12d). This accounts for the presence of derivative topologies, a subset of incorrectly classified molecules within the automated algorithm topology classification. The differing probabilities between the theoretical and observed values are due to the clustering of crossing points where individual crossings are unable to be resolved and thus provide incorrect classifications.

Additionally, the crossing order is more likely to be classified as incorrect for images with lower resolution (Supplementary Fig. 19), or for those with clustered crossings, or close passing DNA duplexes that do not cross. The 5-node twist and torus knots are most highly misclassified, as their crossings cluster together in entropically favourable conformations. The colocalization of crossings increases the configurational freedom of the rest of the chain, and hence increases

entropy[89]. We observe more clustering for the 5-node torus compared to the 5-node twist by both AFM (Supplementary Fig. 20), and coarse-grained simulations (Supplementary Fig. 21), which implies that the availability of an entropic gain is different in different types of knots. The coarse-grained simulations indicate that the effect is dependent on the presence of an adsorptive force towards the surface, equivalent to an electrostatic potential. The adsorptive force increases the confinement strength and confinement free energy, which results in increased colocalisation of the crossings. The adsorptive force also induces a change in the shape and appearance of the knots, which tend to create rosette-like conformations (Supplementary Fig. 21e). Interestingly, the differences between twist and torus 5-node knots are less pronounced in the absence of the adsorptive force (Supplementary Fig. 21c, d)

## Conformational variability is enhanced by surface immobilisation

For catenated molecules in particular, we observed a greater variability in conformation than predicted (Supplementary Figs. 22–23). We would expect the nicked 4-node catenanes to deposit on the mica surface predominantly in the open conformation, due to the expectation that they would have no writhe and are ~1 kb in length, which should minimise the chance of self-crossings. However, after immobilisation on the mica surface for AFM visualisation, a much greater range of conformations is observed, which we categorise as "open" (Fig. 5a), "taut" (Fig. 5b), "clustered" (Fig. 5c) or "bow-tie" (Fig. 5d) conformations. We have interpreted these conformations in schematic

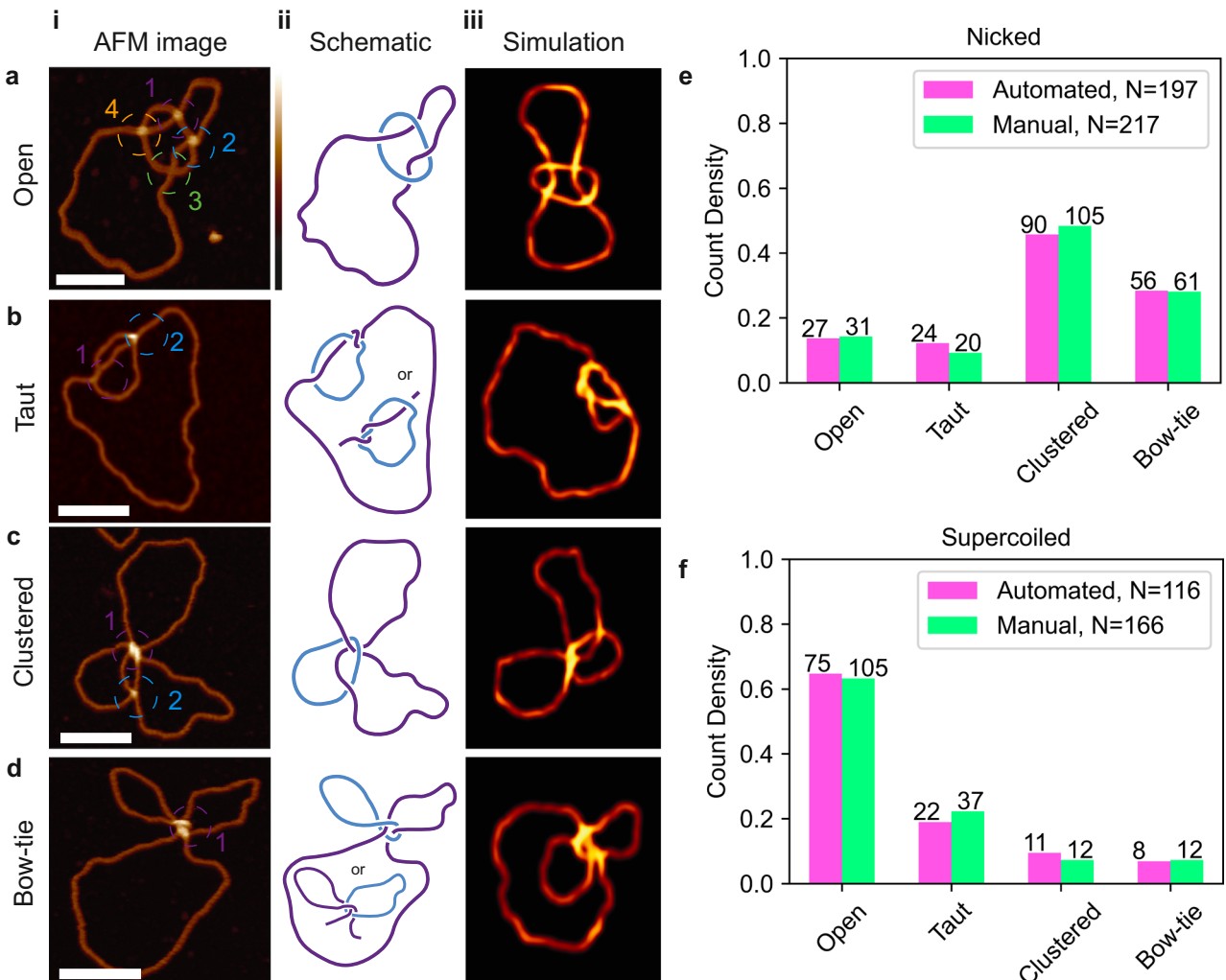

**Fig. 5 | Variability in catenane conformation is driven by topological state.** Conformation classifications of individual molecules from the AFM image are separated into (**a**) open ($N = 136$) (**b**) taut ($N = 57$) (**c**) clustered ($N = 117$) **d** and bow-tie ($N = 73$). Each molecule **i**, is identified using classification rules determined by Supplementary Table 2. **ii** shows representative schematics of each class of the catenane and **iii**, representative pseudo-AFM data from coarse-grained simulations.

Conformational classification of molecules is performed on catenated species in (**e**), nicked (automated, $N = 197$, manual $N = 217$) and **f**, supercoiled (Automated, $N = 116$, Manual, $N = 166$) forms using automated (pink) and manual (green) classification. Scale bars: 50 nm. Height scale: −2 to 6 nm. Source data is provided as a Source Data file.

line diagrams to accommodate the necessary topology, to the right of each respective AFM images (Fig. 5a–d(ii)). The "taut" and "bow-tie" classes are difficult to interpret due to the tight clustering of several nodes, therefore, each schematic diagram shows two possible conformations that accommodate the geometry of the circles and their four catenated crossings.

These conformations make automated topological tracing challenging due to complex crossing architectures and close-passing DNA duplexes. For example, the "bow-tie" (Fig. 4d) crossing can be identified as an open-circle as the single crossing in the molecule is traced through multiple times along the best aligned segments. We develop an automated classifier that enables us to categorise catenanes that could not be topologically traced by their conformation. This classifier calculates the number of nodes and crossing branches for each molecule and assigns it as open, taut, clustered, bow-tie or unclassified, as defined in Supplementary Table 2.

We compare the conformations of nicked and negatively supercoiled 4-node catenanes immobilised using the same magnesium deposition process to see if the DNA mechanics influence the DNA conformation. There is a change in prevalent conformation between the nicked and negatively supercoiled species as determined by both

automated and manual classification. The nicked population adopts a wider range of conformations with a preference for the clustered conformation, whereas the supercoiled presents with a much stronger preference for the open conformation (Fig. 5e, f).

This difference can be explained if the clustered and bow-tie conformations arise from deposition of catenanes where the large circle contains a positive supercoil encircled by the small circle. This positively writhed conformation of the large circle should be strongly disfavoured in the negatively supercoiled catenanes. In contrast, there should be no energetic barrier to the large circle adopting a positively writhed conformation in the nicked catenanes. Indeed, positive writhe induced by the wrapping of the large circle around the small circle has been observed experimentally in the nicked Xer catenane[71]. To better understand the physical properties that drive the DNA to adopt these conformations upon deposition, we perform coarse-grained simulations of the 4-node catenane (Fig. 5a-d(iii)).

**Coarse-grained simulations reveal that twist-writhe partitioning is altered by surface immobilisation**

To understand how surface immobilisation affects the DNA conformation upon deposition in divalent ions, we employ coarse-grained

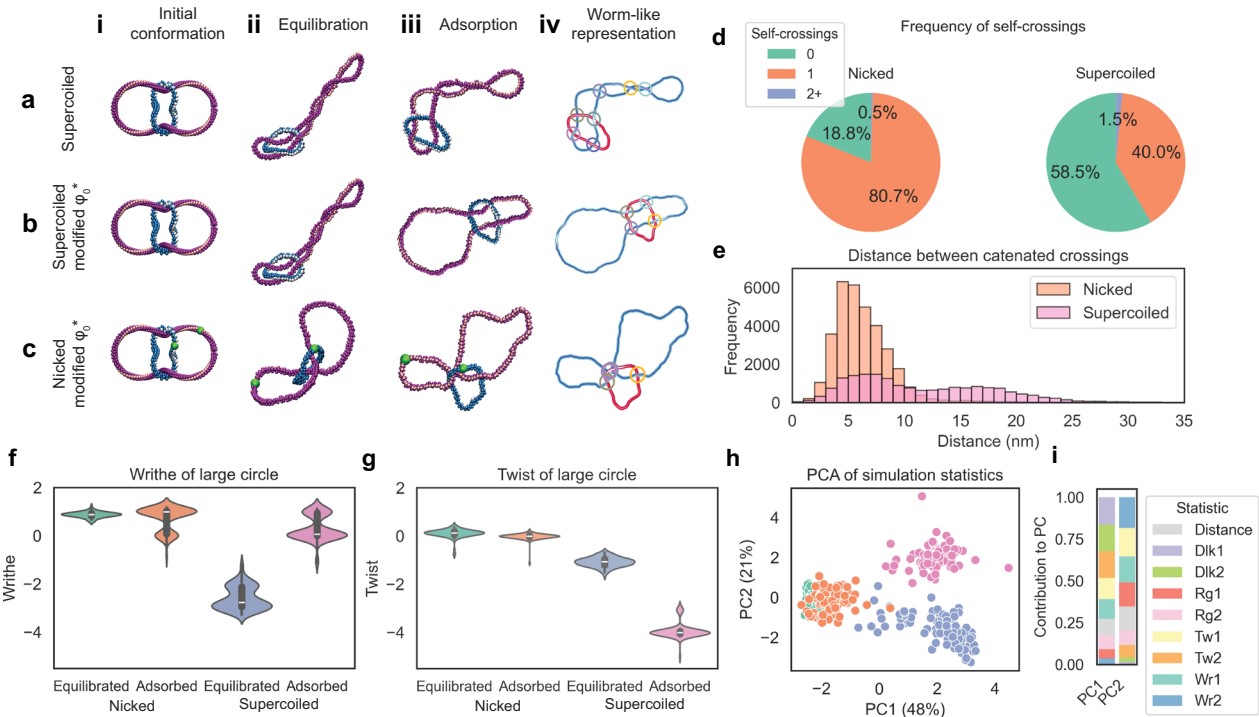

**Fig. 6 | Surface immobilisation drives conformational changes, however, topological species can still be separated by their linking number difference.** Coarse-grained molecular dynamics simulations of catenanes in (**a**) supercoiled with dihedral angle $\varphi_0^* = 0$ (**b**) supercoiled with adjusted dihedral angle and **c** nicked with adjusted dihedral angle in i, initial; ii, equilibrated; iii, adsorbed states and; iv, worm-like representations with crossings between the catenated molecules highlighted. Note that for the remaining sub-figures, both "nicked" and "supercoiled" correspond to simulations with adjusted dihedral angle. **d** Frequency of crossings between the same molecule of DNA within adsorbed catenanes for nicked and supercoiled molecules show a reduction in the number of crossings for supercoiled molecules. **e** Separation of crossings that occur between the two molecules within the catenane for nicked and supercoiled species. **f** Writhe and **g**, twist of the large circle for nicked and supercoiled species in equilibrated and

adsorbed conformations. $N = 100$ Nicked equilibrated, $N = 91$, Nicked adsorption. $N = 100$ Supercoiled equilibration, $N = 85$ Supercoiled adsorption. Summary statistics (min, Q1, Q2, Q3, max, mean) for writhe: Nicked equilibrated (0.56, 0.82, 0.875, 0.94, 1.24, 0.874); Nicked adsorption (−0.9, 0.077, 0.99, 1.03, 1.18, 0.727); Supercoiled equilibrated (−3.27, −2.955, −2.755, −2.125, −1.18, −2.589); Supercoiled adsorption (−1.14, 0, 0.065, 0.94, 1.12, 0.306). For twist: Nicked equilibrated (−0.78, 0.057, 0.13, 0.185, 0.42, 0.107); Nicked adsorption (−1.12, −0.045, −0.010, 0.025, 0.19, −0.025); Supercoiled equilibrated (−1.65, −1.22, −1.065, −0.948, −0.68, −1.082); Supercoiled adsorption (−5, −4.05, −4, −3.95, −2.96, −3.937). **h**, PCA scores plot showing separation of nicked (green for equilibrated, orange for adsorbed) and supercoiled (blue for equilibrated, pink for adsorbed) species on one axis (PC1) and equilibrated and absorbed conformations on the other (PC2). **i**, PCA loadings for PC1 and PC2. Source data are provided as a Source Data file.

molecular dynamics simulations of the 4-node catenane, in both nicked and supercoiled states. We use the conformations obtained from AFM imaging to parameterise our model and reproduce the experimentally observed images (Fig. 5a–d(iii)). To obtain these conformations, we immobilise the DNA in a buffer solution containing magnesium ions, which facilitates weaker binding to the mica surface compared to nickel[76,90], and stiffens the DNA helix[77,91,92]. This immobilisation method allows the molecules to sit in a more open state than is achieved using nickel ions, improving topological classification. AFM measurements also indicate fewer supercoils than expected for the supercoiled catenanes when using this method (Supplementary Fig. 7).

Given native supercoiling of the original DNA substrate and the assumed +4 linkage change of the Xer recombination reaction, the large DNA circle is expected to be underwound by ~4 turns. It has been reported that alternative DNA conformations may be observed at some salt concentrations[59] and that negatively supercoiled plasmids can lose 70–80% of plectonemic supercoils under conditions which allow for dynamics and equilibration on the surface[93]. We hypothesise that the DNA helicity may change during deposition. For our models, we reduce underwinding by altering the number of turns defined by the equilibrium dihedral angle, $\phi \leq 2\pi\Delta Lk/N$[94]. This enables the coarse-grained simulations to achieve more open conformations (Fig. 6b, c) than with the non-adjusted dihedral angle (Fig. 6a), consistent with our AFM experiments. Simulated supercoiled molecules show much more open conformations, as observed in both AFM images and analyses

(Fig. 6b). As discussed above, in nicked catenanes, catenation of the small circle around the larger circle can induce a positive writhe in the larger circle, leading to an increased observation of clustered conformations with the larger circle folded upon itself (Fig. 6c).

As open conformations can be characterised by a lack of proximity in catenated crossings or a decrease in the number of self-crossings, we look to confirm this finding by characterising the frequency of self-crossings (Fig. 6d), and distance between crossings (Fig. 6e). For the supercoiled species, the number of molecules with no crossings is three times that of the nicked molecules, confirming the experimentally observed increase in open conformations. For nicked molecules, catenated crossings are observed in closer proximity to one another, with the distribution of distances between catenated crossings forming a single peak with mean and variance of 6 nm, consistent with "clustered" "taut" and "bow-tie" conformations (Fig. 6e). However, in supercoiled molecules we see a bimodal distribution of catenated crossings, with the first peak sharing the same mean and variance as for nicked molecules, and the second peak having a larger mean (~16 nm) and variance (~19 nm), more consistent with the "open" conformation (Fig. 5e).

For supercoiled molecules, the presence of attractive forces on the surface drives the observed conformations to lower writhe states. This can be quantified by the change in average writhe of the large circle from −2.59 ± 0.05 to 0.31 ± 0.05 following adsorption, indicating a large number of adsorbed conformations are free from self-crossings

(Fig. 6f). The nicked molecules changed little upon surface adsorption with writhe in the larger circle $0.87 \pm 0.01$ at equilibrium and $0.73 \pm 0.05$ upon adsorption (Fig. 6f). The change in writhe of supercoiled species upon adsorption is compensated for by a reduction in twist, with mean twist of the large circle changing from $-1.08 \pm 0.02$ to $-3.94 \pm 0.03$ following surface adsorption (Fig. 6g).

To determine which variables drive the differing behaviours of supercoiled and nicked molecules upon deposition, we perform Principal Component Analysis (PCA) on several statistics extracted from simulations including writhe, twist, linking number difference and radius of gyration, as well as a measure of distance between the centre of mass of small and large catenated circles (Fig. 6h). Through PCA, we achieve good separation between nicked and supercoiled molecules, and also between the equilibrated and adsorbed species of supercoiled, but much less for nicked species. For PC1, we observe that the linking number difference for the large circle and smaller circle have the biggest loading value, indicating that these measures have the greatest contribution to the variance observed between nicked and supercoiled species (0.46 and 0.44, respectively). For PC2 the writhe of the small circle, twist and writhe of the large circle, alongside radius of gyration and distance between the centre of mass of the two circles all have effect (0.49, 0.45, 0.41, 0.38, 0.37) (Fig. 6i). Statistical distributions of individual metrics calculated from simulations are provided in Supplementary Table 4 and Supplementary Fig. 24, while Supplementary Fig. 25 illustrates how these metrics influence the conformation of DNA.

## Discussion

Atomic Force Microscopy (AFM) imaging has transformed our ability to visualise DNA in aqueous solution, offering high-resolution imaging, albeit via immobilisation on a surface, without the need for protein coating[42,43]. We have developed a pipeline which enables us to determine the crossing order, i.e., the over- and under-passing strand for each molecule, enabling tracing and topological classification of complex DNA structures, including plasmids, knots, catenanes and replication intermediates. Our method enables us to determine contour length across DNA species and infer levels of writhe or supercoiling (Fig. 1). We show that we can use these metrics to recover a 56:44 classification of a 50:50 mixed population of nicked and supercoiled DNA plasmids using random forest classification (Supplementary Fig. 9), highlighting the wide-scale applicability of our approach.

However, automating the tracing of topologically complex DNA molecules from AFM images isn't without its challenges, particularly in tracing close-passing or overlapping strands. We address these challenges by developing novel methods for segmentation, tracing, identification of crossings, determination of crossing order, and topological determination for single DNA molecules in a high-throughput and transparent manner. We can identify the over- and under-passing strand even in crossings that show minimal height variation using the full width at half maximum for each crossing segment and determine a pseudo-confidence to inform downstream classification. This integrated pipeline represents a significant advancement in the field, facilitating the identification of complex DNA structures that are difficult to resolve using traditional methods[13,34,35] (Fig. 2).

We apply our pipeline to analyse the structure of DNA plasmids, incubated in nucleoplasmic *Xenopus* egg extract, which is a mix of replicated and unreplicated plasmids and theta structures. We show that we can identify replication forks as 3-way junctions and quantify the length of DNA between them, quantifying the lengths of replicated and unreplicated DNA, and additional features such as reversed forks (Fig. 3).

By studying the Xer recombination system of *E. coli*[18,70,95], we show our method can determine the topology of individual DNA molecules, successfully differentiating between different knot types of the same node number, which are challenging to distinguish by gel electrophoresis. Deriving a 3D topological classification from a topographic image has its challenges, even by eye. As a result, we developed a data cleaning pipeline to remove molecules with clustered crossings from the topological classification analysis, where it was not possible to define a crossing order within the clustered crossing region and accurately extract crossing information from topography. To improve topological determination, each crossing in the automated trace is assigned a reliability score and the crossing order of singular low-reliability crossings can be inverted. By manually calculating all the possible sign conformations of these AFM-traced molecules, we observe that there were no instances of mis-identifying a twist-knot as a torus-knot, or vice versa (Fig. 4).

Additionally, we observe conformational variability in single topological species, indicating the impact of surface immobilisation and how this varies in different topological species. We uncover unexpected conformational preferences in nicked and negatively supercoiled catenanes, using both manual and automated analysis. These conformational preferences may arise from catenation of the small circle around the larger circle, which likely induces positive writhe in nicked catenanes. Conversely, negative supercoiling in negatively supercoiled samples discourages positive supercoiling, resulting in fewer clustered molecules and more open conformations (Fig. 5).

To determine the driving forces behind the changes in confirmation we observe for supercoiled and nicked molecules, we use coarse-grained simulations. We parameterise these using the AFM experimental data and determine that a change to the twist: writhe partitioning of the DNA molecules is necessary to achieve the "open" conformations obtained experimentally. However, the simulations also demonstrate that despite changes to twist, writhe, radius of gyration and molecular separation on adsorption of DNA molecules to a surface, the biggest discriminating factor between the nicked and supercoiled species is still linking number difference (Fig. 6).

Our automated pipeline can identify, trace and classify individual topologically complex DNA molecules across a variety of different unknotted, knotted, catenated and partially replicated plasmids to obtain quantitative and descriptive metrics. Ideal performance is achieved with resolutions between 0.5 and 1 nm/pixel (Supplementary Fig. 19) and when all crossing points are visible and assigned with high reliability, leading to precise contour length estimations (within 7% of predicted length). Though it is theoretically possible to segment and separate molecules that have adsorbed on top of one another, we did not observe these due to the concentration at which this study was performed. The ability to separate overlapping molecules will depend on the orientation of the molecules at the point where they overlap. If overlapping segments are nominally perpendicular to one another, these can be paired effectively and the individual molecules separated.

We observe conformational variation within topological classes, induced by DNA deposition, entanglement, electrostatic interactions and individual flexible sites. Immobilisation conditions must be altered depending on what structures are being observed, and what metrics are desired. In order to determine the supercoiling of small plasmids, $NiCl_2$ immobilisation provides a more representative measure of writhe. When attempting to determine the topology of larger or more complex structures, $MgCl_2$ should be used, as it will create more open conformations with fully separated crossings, improving the ability to trace the molecule.

Our pipeline can be applied to a range of DNA and RNA structures, including those interacting with proteins, and opens avenues for understanding fundamental biological processes which are regulated by or affect DNA topology. Beyond nucleic acids, this approach has wider applications in structural biology, from fibrillar structures to polymeric networks and could inform further advances in DNA nanotechnology and structural biology.

## Methods

Strains used throughout this work can be found in Table 1.

## Protein purification

PepA was overexpressed in an *argR⁻ E. coli* strain DS956. PepA cell pellets were resuspended in a lysis buffer [50 mM Tris pH 8, 1 M NaCl, 1 mM EDTA] containing protease inhibitor cocktail (cOmplete™ Mini, Roche) and lysed by sonication. The lysate supernatant was treated with a 70 °C heating step for 10 minutes, where PepA remains soluble, and denatured proteins were then removed by centrifugation at 40,000 g for 20 minutes at 4 °C. PepA was then further crudely purified by ammonium sulphate precipitation to 50% saturation (0.291 g/ml at 0 °C). The ammonium sulphate precipitation pellet was resuspended using a resuspension buffer [20 mM Tris pH 8, 200 mM NaCl, 1 mM EDTA, 2 mM DTT], then dialysed overnight (4 °C) against a low ionic strength dialysis buffer [20 mM Tris pH 8, 10 mM NaCl, 1 mM EDTA, 2 mM DTT], causing precipitation of the PepA within the dialysis tubing. PepA was then further purified by two FPLC (Fast Liquid Protein Chromatography) steps. The precipitated PepA was collected by centrifugation at 40,000 g for 20 minutes at 4 °C, then resuspended with PepA Buffer C [20 mM Tris pH 8, 200 mM NaCl, 1 mM EDTA, 10% Glycerol]. The crude PepA solution was loaded onto a HiTrap Heparin-HP 1 ml column (GE Healthcare), then washed with PepA Buffer A [PepA Buffer C without Glycerol]. A gradient elution was performed from PepA Buffer A to PepA Buffer B [20 mM Tris pH 8, 2 M NaCl, 1 mM EDTA]. Elution fractions were combined into a single pool that was subsequently adjusted to conditions equal to that of PepA Buffer D [20 mM Tris pH 8, 1 M (NH₄)₂SO₄, 1 M NaCl, 1 mM EDTA]. The adjusted pool was loaded onto a 1 ml Phenyl-HP hydrophobic interaction column (GE Healthcare) pre-equilibrated with PepA Buffer D then washed with the same buffer. Protein was eluted during a gradient from PepA Buffer D to PepA Buffer E [20 mM Tris pH 8, 1 M NaCl, 1 mM EDTA]. Peak absorbance eluted fractions were pooled and dialysed against PepA storage buffer [50 mM Tris pH 8, 1 M NaCl, 1 mM EDTA, 2 mM TCEP, 50% glycerol] and stored at −20 °C.

ArgR was overexpressed in a *pepA⁻ E. coli* strain DS957 and purified essentially following the procedure of Burke et al.,[96] but used FPLC steps of heparin-affinity using a HiTrap Heparin-HP 1 ml column (GE Healthcare), followed by anion exchange chromatography using a MonoQ 5/50GL (GE Healthcare) column. ArgR was bound to each column using "2xTM" buffer + 50 mM NaCl [20 mM Tris pH 7.5, 20 mM MgCl₂, 2 mM DTT, 50 mM NaCl], and then eluted from each column using "2xTM" buffer + 1 M NaCl. MonoQ-purified pools of ArgR were dialysed to ArgR storage buffer [10 mM Tris pH 7.5, 10 mM MgCl₂, 75 mM NaCl, 2 mM TCEP, 50% Glycerol], and stored at 20 °C.

An MBP-XerC fusion protein was overexpressed in a *xerC⁻ xerD⁻ E. coli* strain DS9029. The cells were resuspended in MBP lysis buffer [20 mM Tris pH 8, 200 mM NaCl, 1 mM EDTA, 2 mM DTT] and lysed by sonication. FPLC was performed by loading the lysate supernatant onto a 1 ml MBP-Trap column (GE Healthcare), followed by a wash step using the same lysis buffer, and then elution with the same buffer containing 10 mM maltose. Pooled elution fractions were subjected to a 2nd FPLC stage using a 1 ml HiTrap Heparin-HP 1 ml column (GE Healthcare). The bound MBP-XerC was washed using heparin loading buffer [50 mM Tris pH 8, 200 mM NaCl, 1 mM EDTA, 2 mM DTT, 10% glycerol], then eluted by a gradient to the same buffer containing 1 M NaCl. Peak elution fractions were pooled and dialysed against a storage buffer [50 mM Tris pH 8, 1 M NaCl, 1 mM EDTA, 2 mM TCEP, 50% Glycerol] and stored at −20 °C. MBP-XerC was used as an active recombinase without removal of the solubility tag.

XerD was overexpressed in *xerC⁻ xerD⁻* strain DS9029 and purified by native (non-His-tagged) nickel-affinity FPLC as performed in Subramanya et al.[97], using a 1 ml HisTrap-HP column (GE Healthcare), then by a Heparin-affinity purification using a 1 ml HiTrap Heparin-HP column (GE Healthcare). Other FPLC steps from Subramanya et al. were omitted. Enriched Heparin-affinity elution fractions of XerD were pooled and dialysed against a storage buffer [50 mM Tris pH 8, 1 M NaCl, 1 mM EDTA, 2 mM TCEP, 50% Glycerol] and stored at −20 °C.

## Creating a miniaturised Xer catenation substrate with uneven product circles

The 3406 bp plasmid pSDC153[72] contains two directly repeated end-to-end *psi* recombination sites. This *psi*-site doublet was excised from pSDC153 by restriction digest using flanking restriction enzymes EcoRI and HindIII, and ligated into the 885 bp microplasmid vector piAN7[95] (Supplementary Fig. 1b). piAN7 contains the minimal DNA sequences required for plasmid replication by utilising a *supF* amber-suppressor tRNA selection strategy. For selection, piAN7 must be paired with an amber-codon mutagenized ampicillin/tetracycline resistance plasmid "p3" (60 kb). The newly miniaturised *psi* recombination substrate, p4CAT, is 1651 bp (Supplementary Fig. 1c). As piAN7-based plasmids rely on amber-suppression for antibiotic selection, the host strain must have a *sup⁰* genotype. Empty piAN7 vector was maintained in *sup⁰* strain HMS174 and p4CAT was maintained in *sup⁰ xerC⁻* strain DS953*xerC-y17*. Amber suppressor selection plasmid p3 was freshly conjugated into both of these strains prior to use. piAN7 and derivatives were maintained by antibiotic selection on solid or liquid media with 25 μg/ml ampicillin and 10 μg/ml tetracycline. Xer recombination reactions of p4CAT were analysed by restriction digest with various restriction endonucleases (HindIII or BamHI) and/or nickases (Nb.BssSI or Nt.BsmAI) to reveal the relative mobility of the constituent circles within the p4CAT catenane under supercoiled/nicked/linear conditions by gel electrophoresis (Supplementary Fig. 1d).

## Bulk DNA substrate preparation

Large preparations of p4CAT DNA were made using a QIAGEN Maxiprep kit. p3 amber-selection "helper" plasmid DNA was selectively removed from plasmid preparations by a 1-pot endo- and exonuclease digestion reaction protocol adapted from (Balagurumoorthy et al.)[98]. Such digestion reactions contained EagI-HF restriction endonuclease for multi-linearization of p3, along with lambda-exonuclease and recJ^F-exonucleases for 5"-3" dsDNA and 5"-3" ssDNA digestion, respectively. A typical large preparation of substrate contained roughly 200 ug DNA, incubated in 1x NEB Cutsmart buffer conditions for 20 hours at 37 °C with 50U of EagI-HF, 30U of Lambda-exonuclease, and 260U of

**Table 1 | *E. coli* strains used during this work and their associated origin**

| Strain | Genotype | Reference |
|---|---|---|
| DS941 | *E. coli* K12 AB1157 *recF143 lacIq lacZΔM15* (StrR) | (Summers and Sherratt)[113] |
| DS956 | DS941 *argR::folA* (StrR TpR) | (Flinn et al)[114] |
| DS957 | DS941 *pepA::Tn5* (StrR KanR) | (Stirling et al)[95] |
| DS9029 | DS941 *xerC::CmR xerD::fol* (StrR CmR TmR) | (Colloms)[18] |
| HMS174 | *E. coli* K12 F⁻ *recA1 hsdR⁻K⁺* (RifR) | (Campbell et al)[115] |
| DS953 *xerC-y17* | DS941 *sup⁰ xerC::CmR* (StrR CmR) | This work |
| JAM103 | *E. coli* K12 SURE (Agilent; *recB recJ sbcC umuC uvrC) xerC::GentR* (KanR, TetR, GmR) | This work |
| FC33 | DS941 *xerC::KanR recA::TetR* (StrR KanR TetR) | This work |

recJ[F]. Enzymes were removed from the reaction products by extraction with phenol:chloroform:isoamylalcohol (25:24:1), followed by two treatments of the aqueous phase with chloroform:isoamylalcohol (24:1) to remove traces of phenol. The product DNA was ethanol precipitated and resuspended in TE buffer.

## Assembly of pDIRECT/pINVERT

pKS492 and pKS493[99] each contain a different orientation of the 280 bp, HpaI-TaqI "cer-site" fragment from the ColE1 natural plasmid, within the pUC18 cloning vector MCS[100]. Restriction-digested fragments of pKS492, pKS493, and a PCR amplified chloramphenicol resistance (cmR) were combined in a 3-part DNA ligation to assemble a cer site flanking either side of cmR in a pUC18 vector. This generated four new plasmids containing the four different combinations of cer site orientations with respect to the fixed orientation of cmR. Subsequently, the HindIII-EcoRI restriction fragments containing cer-cmR-cer in the direct- or inverted-repeat site orientations were cloned into piAN7, generating the 2280 bp plasmids pDIRECT and pINVERT. Due to the cmR marker, it was not necessary to maintain these plasmids in sup⁰p3 E. coli strains. pDIRECT was cloned and amplified in strain FC33 without issue. Due to its large inverted repeats, pINVERT had to be cloned and amplified in JAM103, an xerC derivative of the commercially available strain SURE (Agilent), which tolerates inverted-repeat DNA. Large quantities of contaminating gDNA in JAM103-derived DNA preparations were removed using the same 1-pot endo-exonuclease reaction described above.

## DNA knot pINVERT substrate preparation

DNA preparations of pINVERT from JAM103 contained approximately <10% multimeric intermolecular plasmid recombination products. Upon linearisation with HindIII these multimers yield incorrect DNA orientations, and therefore contaminate the linear DNA in downstream ligation-knotting reactions. Homogenous pINVERT of the correct type was purified by large-scale low-melting point gel-electrophoresis of supercoiled pINVERT plasmid, followed by excision of the correct monomeric supercoiled species. This DNA was then extracted from the low-melt agarose. The homogenous supercoiled pINVERT was then linearised with HindIII and subjected for experiments.

## Xer recombination reactions

In vitro Xer recombination reactions were performed in a buffer containing 50 mM Tris-HCl pH 8, 25 mM KCl, 1.25 mM EDTA, 5 mM spermidine, 25 μg/ml BSA, and 10% glycerol. Supercoiled plasmid DNA was added to a final concentration of 21 nM. Proteins were added to final concentrations of ~300 nM PepA (Hexamer concentration) and ~250 nM XerC and XerD (monomers). Reactions were incubated at 37 °C for 60 minutes, then the DNA was purified from the reaction components by phenol:chloroform:isoamyl extraction (25:24:1), followed by two treatments with chloroform:isoamyl, then ethanol precipitation and resuspension in TE buffer. Restriction digests and nicking-endonuclease reactions on purified recombinant DNA were carried out as per the manufacturer's instructions.

## DNA catenane purification for AFM

Large quantities of DNA catenanes were homogeneously purified away from un-recombined substrate DNA by agarose gel extraction. Extractions of supercoiled catenanes used the QIAGEN gel extraction kit as per the manufacturer's instructions. Nicked catenanes (treated with Nt.BsmAI - singly nicking both circles) were extracted using 1% low-melting point agarose (SeaPlaque GTG, Lonza) gels using β-Agarase I (NEB) digestion. Low-melting point gel slices were weighed and an equal volume of nuclease-free water was added. 10x β-Agarase I Buffer was added to equal 1x final concentration. The gel slice was then crushed into a paste with a glass rod, melted at 65 °C for 10 minutes with repeated vortexing, and cooled to 42 °C. 2U of β-Agarase I enzyme was finally added per 200 mg of gel-slice, and the

digestion reaction incubated at 42 °C for 16 hours. Agarase reactions were centrifuged at 12,000 g for 10 minutes and their supernatants were transferred to fresh tubes. The extracted DNA supernatants were further purified by the phenol:chloroform-ethanol precipitation steps and resuspended in TE buffer for use. The homogeneity of these gel-extracted samples prior to AFM imaging was observed by gel electrophoresis (Supplementary Fig. 1e).

## DNA knot generation

DNA knots were generated by circularisation of linear DNA substrates with T4 DNA ligase in the presence of Xer accessory proteins PepA and ArgR, approximately following the procedure described by Alén et al. [72]. Each reaction contained 17 nM linearised substrate DNA (pDIR or pINV) with 250 nM PepA (hexamer) and 30 nm ArgR (hexamer) in a reaction buffer condition (1 x "cer buffer") of 50 mM Tris HCl pH 7.5, 25 mM NaCl, 1 mM L-arginine, 25 μg/ml BSA, 2.5 mM spermidine, 2.5 mM DTT. Reaction mixtures were then incubated at 37 °C for 15 minutes for nucleoprotein synapse formation. This was followed by addition of "ligase mix" ("cer buffer" plus T4 DNA ligase, 30 mM MgCl₂, 3 mM ATP), which leads to a final concentrations of 50 units T4 DNA ligase, 10 mM MgCl₂, and 1 mM ATP. DNA was ligated for 60 minutes at room temperature, then reactions were heat-inactivated at 65 °C for 15 minutes. Finally, the reaction products were nicked to remove supercoiling by incubation at 37 °C with Nt.BsmAI or Nb.BssSI nicking endonucleases (NEB) within the existing reaction mixture. For AFM analysis, the mixed knot production reactions were scaled-up and finished with DNA purification by phenol:chloroform:isoamyl extraction (25:24:1), then two extractions with chloroform:isoamyl alcohol (24:1), followed by ethanol precipitation and resuspension in TE buffer. Single knot types (e.g., 5-torus, or 5-twist alone) were isolated by the separation of nicked knot reactions in 1% low-melting point agarose gels (SeaPlaque agarose, Lonza), followed by excision of specific bands. DNA extractions from the agarose slices used β-agarase (NEB) enzymatic digestions as per the manufacturer's instructions. Crude β-agarase extracted DNA was further purified for AFM by phenol:chloroform:isoamyl extraction (25:24:1), then two extractions with chloroform:isoamyl alcohol (24:1), followed by ethanol precipitation and resuspension in TE buffer.

## DNA electrophoresis

For topological DNA electrophoresis, 1% agarose gels were used with a modified TAE running buffer "TSAE" [40 mM Tris acetate, 20 mM sodium acetate, 1 mM EDTA] using 23 × 16 cm gel electrophoresis kits, for 16–24 hours (1.5 V/cm). Gels were stained with SYBR® Gold DNA stain and laser-scanned using a GE Healthcare Typhoon FLA9500 laser scanner at 450 V PMT on the SYBR® Gold preset (474 nm; LBP filter). Unprocessed gel images are included with the source data.

## Replication intermediate production from *Xenopus* egg extracts
**Reagents and chemicals.** Creatine phosphate (PC), disodium salt (Sigma), was made up as a 1 M stock dissolved in 0.1 M KHPO₄ and stored at −20 °C as single use aliquots. Creatine phosphokinase (CPK - Roche) was dissolved in a buffer containing 10 mM Hepes (pH 7.5), 50% glycerol and stored at −20 °C. ATP (Cytiva) was prepared as a 180–200 mM aqueous solution at pH 7.5, adjusted with NaOH. 3,4-Dihydroxybenzoic acid (PCA – Sigma) was prepared as a 250 mM solution in water with pH adjusted to 8.0 with KOH. 4,5′,8-Trimethylpsoralen (TMP – Sigma) was dissolved in EtOH at 1 mg/mL and stored in the dark at −20 °C. Phenol:chloroform (Sigma), Chloroform.

**DNA substrates.** pJD97 – Lac48 was a gift from Professor James Dewar. pBS SK + (Agilent). pET11a-LacI-bio was a gift from Dr Kenneth Marians (MSKCC). pUCattB-Ter24 was digested with HindIII and ligated to a 5′-phoshorylated oligonucleotide duplex (IDT) containing 4 TerB sequences to yield pUCattB-Ter4. GGATCCTCACACCTACAA GGGATGTACATCAATTAGTATGTTGTAACTAAAGTGTTAGGGAGGAA

TTAGTATGTTGTAACTAAAGTTGGAGTTGATAATTAGTATGTTGTAAC TAAAGTGGCTTCAACGTAATTAGTATGTTGTAACTAAAGTTCCGTAC GAATGTGCCGAACTTATAAGCTT. The repeat array was excised from pUCattB-Ter4 with BsrGI and BsiWI, gel purified and religated into the cut vector. Colonies were picked and sequenced to yield plasmids with 24 TerB repeats that were verified by sequencing and restriction digest mapping.

**StrepII-Tus expression and purification.** The protein sequence (*E. coli* | EG11038) of Tus was fused to the following N-terminal tandem Strep-tag II and linker sequence: MGSAWSHPQFEKGGGSGGGS GGSAWSHPQFEKGGGS. DNA sequences were codon optimised for *E. coli* and synthesised to be in frame with pET28a by Twist Biosciences. 4 L of BL21(DE3) transformed with pET28a-Strep_Tus were grown in 2XYT broth containing kanamycin (30 µg/mL) at 37 °C until an OD of 0.5-0.7 was reached and protein induction achieved by the addition of IPTG to 1 mM. Growth was continued for 3 hrs at 37 °C and cells harvested by centrifugation at 31,000 *g*. Cell pellets were resuspended in a buffer containing 50 mM tris.Cl (pH 7.5) and 10% sucrose and stored at −80 °C. Cell suspensions were thawed and 4 EDTA-free protease inhibitor tablets (Roche), PMSF (0.1 mM), EDTA (0.1 mM) and DTT (1 mM). Cells were lysed on ice using a Branson 450 sonifier at 70% power and 50% duty for 4 × 1 min with intervening rests on ice-water for 5 mins. The lysate was made up to 500 mM NaCl and clarified by centrifugation at 20,000 RPM, 31,000 *g* in a JA 25.50 rotor for 30 mins at 4 °C. Polymin P was added to the lysate to 0.4% with stirring at 4 °C for 15 mins. Nucleic acids were pelleted at 10,000 RPM for 20 mins (JA 25.50) at 4 °C. The supernatant was removed and precipitated with solid ammonium sulphate that was added slowly to 80% saturation (0.53 g/mL) at 4 °C. The suspension was stirred for 30 mins and then spun at 20,000 RPM (JA 25.50) for 30 mins at 4 °C. The pellet was resuspended in 40 mL TED25 buffer, 0.2 µm filtered and applied to 2 × 5 mL StrepTrap XT (Cytiva) columns run in series at 1 mL/min at 4 °C. The columns were washed at 1 mL/min until UV reached baseline and proteins eluted with 60 mL TED150 + 50 mM biotin. Peak fractions were pooled and dialysed overnight against the TED75 buffer at 4 °C. The protein solution, with slight precipitation was 0.2 µm filtered and applied to a MonoQ 5/50 GL column at 1 mL/min that was equilibrated in TED100 buffer at 4 °C. Tus protein was collected from the flowthrough and precipitated by addition of solid ammonium sulphate to 80% saturation with stirring for 30 mins. Protein was recovered by centrifugation at 20,000 RPM (JA 25.50) for 30 mins and the pellet resuspended in TED100 + 25% glycerol. Protein was dialysed overnight at 4 °C against the same buffer. Protein concentration was determined using an extinction coefficient of 50482.5 M/cm.

**bioLacI expression and purification.** 4 L of BL21(DE3)-pBirACm transformed with pET11a-LacI-bio were grown in LB broth containing ampicillin (100 µg/mL) and chloramphenicol (34 µg/mL) at 37 °C. Protein expression was induced at an OD of 0.5−0.6 with the addition of IPTG and d-biotin to 1 mM and 50 µM, respectively, and growth continued for 3hrs at 37 °C. Cells were harvested by centrifugation at 31,000 g and pellets resuspended in 50 mM tris pH 7.5, 100 mM NaCl 10% sucrose and stored at −80 °C.

Cells were gently thawed and supplemented with 4 EDTA-free protease inhibitor tablets (Roche), PMSF (0.1 mM) and DTT (1 mM). Cells were lysed on ice using a Branson 450 sonifier at 70% power and 50% duty for 4 × 1 min with intervening rests on ice-water for 5 mins. The lysate was clarified by centrifugation at 20,000 RPM, 31,000 g in a JA 25.50 rotor for 30 mins. The supernatant was made 35% saturated with ammonium sulphate (0.21 g/mL) and stirred at 4 °C for 30 mins. The suspension was spun at 20,000 RPM (JA 25.50) for 30 mins. The pellet was resuspended in a buffer containing 50 mM tris.Cl - pH 7.5,

0.1 mM EDTA, 1 mM DTT and 100 mM NaCl – abbreviated as TEDX, where X denotes NaCl concentration. The protein solution was 0.2 µm filtered and applied to a 10 mL softlink avidin (Promega) column that had been equilibrated in TED100 at ~1 ml/min at room temperature. The flow-through was collected and reapplied to the column. The column was washed with 50 ml TED100. Proteins were eluted with 25 mL TED100 + 5 mM biotin. After 30 minutes another 25 mL TED100 + 5 mM biotin elution. Eluates were pooled and dialysed overnight in TED 60 at 4 °C. The protein solution 0.2 µm filtered and loaded onto a 5 mL HiTrap Heparin HP column (Cytiva) at 2 ml/min that was equilibrated with TED60 at 4 °C. The column was washed with TED60 buffer until UV reached a stable plateau. LacI was eluted with a gradient of TED60-TED1000 over 15CV. Peak fractions were pooled and dialysed overnight against TED100 + 10% glycerol. Protein concentration was determined using an extinction coefficient of 28482.5 M/cm.

**Extracts.** HSS (High-Speed Supernatant) and NPE (NucleoPlasmic Extract) egg extracts were made as described in Sparks et al. [101] with the following modifications. All buffers apart from L-cysteine were chilled at 4 °C overnight and moved to room temperature at the start of the procedure. Crude S phase extract was collected by puncturing tubes with a 5 mL syringe connected to an 18 G needle instead of by gravity. Crude S phase extract was diluted with 0.1 vol ELB buffer prior to ultracentrifugation when making HSS.

**Bulk replication reactions.** DNA templates were licensed at a final concentration of 10 ng/µL by mixing with HSS extract that had been supplemented with ATP (2 mM), CPK (5 µM) and PC (20 mM) and incubated at room temperature for 5 mins. Licensing was carried out for 30 minutes. Replication was initiated by adding 2 volumes of NPE that was also supplemented with ATP (2 mM), CPK (5 µM), PC (20 mM), dATP[α-32P] (Perkin Elmer) (1/50 vol) and equilibrated at room temperature for 15−20 mins. Timepoints from the point of NPE addition are denoted in figure legends. At indicated times, 1.5 µL of the reaction mixture was removed and added to 12 volumes of a stop solution to give final concentrations as follows: EDTA (50 mM), Ficoll 400 (5%, w/v), SDS (2.5%, w/v), bromophenol blue (0.0625%, w/v) and proteinase K (1 mg/mL). Reactions were deproteinised for 1−1.5 hrs at 37 °C prior to being run on 1% TAE gels (12×14 cm) at 25 V for 16.5 hrs. The gel was trimmed just above the dye front and sandwiched between a layer of filter paper and Hybond N+ membrane. A stack of paper towels were placed on top of the gel and a ~1.5 Kg weight was placed on top of the gel for 30−45 mins. The squashed gel was then dried under vacuum at 80 °C for 1.5 hours. After drying, the filter papers were removed and the dried gel and attached Hybond membrane was exposed to a storage phosphor screen and imaged using a Typhoon imager (Cytiva). Each NPE preparation was diluted (typically 50-70%) with an amount of LFB1/50 REF buffer that gave maximal rates and extents of DNA replication.

For experiments that induce site-specific fork stalling, LacI (163 pmols) or Tus (279 pmols) proteins were prebound to DNA templates (300 ng) in their storage buffer for 10 mins at room temperature. The entire volume was added to HSS to license the DNA at a final concentration of 10 ng/µL. An appropriate volume of this mixture was removed after 30 mins and added to 2 vols of NPE as described above. Sample processing and gel electrophoresis were performed as described above.

**AFM sample preparation.** Samples for Atomic Force Microscopy were prepared by preparing licensing reactions as described above at 15 ng/µL for 30 mins prior to replication initiation with 2 vols of NPE. Reactions (26 µL) were terminated at indicated times, under conditions that slow branch migration, by the addition of 4 volumes of stop buffer containing: 62.5 mM tris.Cl pH 7.5, 0.625% SDS and 12.5 mM

MgCl$_2$ in order to slow branch migration. TMP was added to 10 μg/mL and samples were crosslinked by irradiation at 340 nm with a UV Stratalinker 3400 for 10 mins. This process was repeated three times in total. RNaseA was added to a final concentration of 1 mg/mL and samples were digested for 3 hrs at 37 °C. Proteinase K was added at 1 mg/mL and samples were incubated for a further 1 hour at 37 °C. Nucleic acids were twice extracted with phenol:chloroform then chloroform before ethanol precipitation. DNA was resuspended in 10 mM tris.Cl pH 8.0. For each DNA species. For mixed population samples, 3 ng of both supercoiled and relaxed unknot pDIRECT plasmid DNA was premixed in an eppendorf tube to make 6 ng total DNA, which was then immobilised as described below with either nickel chloride or magnesium chloride buffer.

**AFM imaging.** Atomic Force Microscopy of DNA samples was performed by depositing ~5 ng of DNA onto discs of freshly cleaved muscovite mica. A ~20 μl droplet of either nickel [20 mM HEPES pH7.4, 3 mM NiCl$_2$] or magnesium [10 mM TRIS, 25 mM MgCl$_2$] adsorption solution was placed on the mica disc, then the 5 ng DNA sample was pipetted into the droplet and mixed thoroughly by further pipetting[60]. The sample disc was then incubated for 30 or 5 minutes for nickel and magnesium solution, respectively, at room temperature in an enclosed container with a humid environment, to reduce evaporation. After incubation, the sample was washed three times to remove any non-adsorbed DNA by the addition of nickel imaging solution [20 mM HEPES pH7.4, 3 mM NiCl$_2$], vigorous pipetting, then removal of the same volume. This was followed by increasing the final volume of the solution droplet to 25 μl with imaging solution prior to imaging. Sample scans were captured using either a Multimode 8 AFM system (Bruker) or a Dimension XR FastScan (Bruker) using PeakForce Tapping mode. One limitation of PeakForce Tapping mode is that it has lower throughput than standard Tapping mode. Despite the lower throughput, PeakForce tapping was used because it allows for accurate force control to prevent damage to DNA and easier quantification of imaging forces, allowing for consistent high-quality imaging using parameters which rarely vary by more than 5%[44,60,62]. For Multimode use, PeakForce HiResB cantilevers (Bruker) and PeakForce Tapping mode were used. Initially, low-resolution scans [e.g., 1–2 μm$^2$ scan size, 3 Hz scan rate, 256 × 256 lines per scan] were used to quickly identify areas of interest. High-resolution scans [e.g,. 0.18-1 μm$^2$ scan sizes, 1.41 Hz scan rate, 512 × 512 lines per scan] of molecules of interest were carried out with fine-tuned operational settings for maximum imaging resolution [20 nm PeakForce amplitude, 4 KHz PeakForce frequency, and PeakForce setpoints in the range of 5–20 mV, corresponding to peak forces of <70 pN]. For Dimension XR FastScan use, FastScan-D cantilevers (Bruker) and PeakForce Tapping mode were used. Images were taken at 512 × 512 pixels at a scan rate of between 4 and 8 Hz for larger scans of areas. High resolution images were taken at 1024 × 1024 pixels at an appropriate image size (0.3–0.7 μm$^2$) scan size to ensure a resolution of >0.75 nm/pixel at a scan rate of 3–4 Hz. For each sample, a minimum of 5 separate immobilisations were performed with multiple different molecules imaged per condition, totalling 400 images. The number of molecules selected for analysis is shown in the caption of the corresponding figure.

**Manual processing of AFM images for manual interpretation and analysis.** The open-source software package Gwyddion[75] was used to pre-process the data according to previously published work[102]. This removed surface tilt via mean plane subtraction, align rows by row medians, correct horizontal scars, obtain a foreground mask, assign a false-coloured topographic scale, apply a Gaussian filter of 3 pixels, and shift the minimum height to zero. The arbitrary line profile extraction tool in Gwyddion was used to determine the height across DNA crossings by hand using an average of 3 parallel lines.

Additionally, this was used to measure the unreplicated length of DNA manually in the replication intermediate structures, as well as the fork length.

**Automated processing of AFM images for interpretation and analysis.** AFM images were processed using the open-source AFM image analysis software TopoStats[79] (https://doi.org/10.15131/shef.data. 22633528.v2). Flattened images were obtained following similar steps to the Gwyddion processing using median row alignment, planar tilt removal, quadratic tilt removal and scar removal. A background mask obtained from pixel heights below 1σ height allowed the above steps to be repeated on just the background data. The image data was translated so the background average was centred around 0. Finally, a 1.1px Gaussian filter reduced any high-gain noise. DNA molecules of interest were identified using the standard TopoStats grain finding pipeline with the parameters found in the provided configuration file. Object bounding boxes used to crop the identified grains for finer segmentation within a U-Net.

**U-Net finer segmentation.** For training, 76 hand-labelled cropped AFM images underwent random augmentations of; up to a 30% scale increase, up to a 30% translation, integer rotations of 90 degrees, and horizontal or vertical flipping. All crops were upscaled or downscaled into the network as 512 × 512 pixels. The labelled images were split into training and test sets using an 80:20 ratio. The model was trained with a learning rate of 0.001, a batch size of 5, 120 steps per epoch, and trained for 100 epochs. The Adam optimiser and binary cross-entropy loss function were used. The U-Net architecture was constructed as a 5-layer encoder-decoder network with skip connections with specific parameters (Supplementary Fig. 26). Using a test dataset of 15 images, the resultant trained model obtained a dice score of 0.82 and intersection over union score of 0.70 on pixel probabilities, but 0.84 and 0.72 using a binary threshold of >0.1 as in the final workflow to prevent breaks in the mask.

**DNA tracing algorithm.** Individual grain masks are skeletonised using our intensity-biased variation of Zhang's skeletonisation algorithm[81]. Terminating branches with less than 15% of the total of the number of skeleton pixels were pruned, along with branches whose centre height was less than the average skeleton height minus 0.85 nm (the depth of the DNA major groove). A sliding window convolution of the pruned skeleton counts the number of pixel neighbours where > 3 neighbours indicates crossings, and 2 neighbours indicate the backbone. Crossing points separated by up to 7 nm (~ double the tip-convolved DNA width) are connected as they likely belong to the same crossing. The shortest path between two nodes with an odd number of emanating branches (resulting from insensitive segmentation) is also connected. Then, for each 20 nm crossing region (below the 45 nm persistence length of DNA[87]), emanating branches are paired based on their bipartite graph maximal matching vectors, and height traces along the pair are used to calculate the FWHM and determine the overlying and underlying branches. To order the trace, the skeleton was split into connecting and crossing regions, and each linear segment ordered, starting from an endpoint or random connecting region and following on from the previous segment until the trace reached its starting point. If all segments were not used, another molecule trace was started, such as for the DNA catenanes. Simplified topological traces used the over/under-passing crossing region classification to label heights as ascending integers.

**Crossing order reliability.** A pseudo-confidence value used to rank the crossing order reliability (COR) for each crossing is obtained using the FWHM from each duplex height trace in the crossing. For a single crossing (which may contain >2 crossing segments), all possible

combinations (not permutations), N, of the calculated FWHM for each crossing segment are paired with another in the same crossing, FWHM_pairs. A confidence value for each pair of crossing duplexes is obtained, and when > 2 crossings, averaged to obtain an average crossing reliability value using Eq. 1.

*Examples.* (Simple) For two crossing duplexes, with FWMHs of 0.2 and 0.5. The possible pairs are just [0.2, 0.5] so $N = 1$ and the COR = 0.6. (Complex) For 3 crossing duplexes, with FWMHs of 0.2 and 0.5 and 0.7. The possible pairs are: [0.2, 0.5], [0.2, 0.7] and [0.5, 0.7] so $N = 3$ and the values within the summations are 0.6, 0.71 and 0.29. The COR is then the average of these so COR = 0.53 for this single crossing.

For multiple crossings, this would only work if branches are paired correctly, and if the crossing is a Reidemeister move and not a clustered crossing.

**Calculating crossing classification probability.** We took AFM images of DNA 3-Node knots ($N = 10$) and 4-Node catenanes (nicked, $N = 5$, supercoiled, $N = 14$) in open configurations with all crossings ($N = 106$) clearly visible. For both the 3-Node knot and 4-Node catenane samples, only one crossing needs to be clearly determined by eye for the ground truth of all crossings to be known due to the sample topology. Once at least one stacking order is accurately determined, the other more ambiguous crossings can be assigned as they must alternate in their stacking order along the strand path for these specific topologies. Using this simple model, the accuracy of the pipeline was calculated at 82% compared to the hand labels.

Provided a probability of obtaining a correctly classified crossing, $p = 0.82$, the probability of obtaining the different possible topological species from reversing the stacking order in each crossing can be calculated.

For example, a 3-node knot topology can be obtained by correctly or incorrectly classifying all crossings of a 3-node knot. Incorrectly classifying just one or two crossings will result in an unknot topology. Combinatorics tells us that there are 6 ways to obtain an unknot (3 with one misclassification and 3 with two crossing misclassifications), and 2 ways to obtain a 3-Node knot (Supplementary Fig. 4). Using the probability above with the combinatorial values, we obtain the probability of correctly classifying complex topologies of increasing crossing numbers (Supplementary Fig. 18).

**Random Forest classification.** To assess the ability of our approach to distinguish between relaxed and supercoiled conformations, we trained a random forest classifier on a data set of Ni immobilised nicked and supercoiled molecules, consisting of 31 relaxed molecules and 49 supercoiled molecules (Supplementary Fig. 9). The model was built using the following list of features: 'smallest_bounding_width', 'smallest_bounding_length', 'aspect_ratio', 'max_feret', 'grain_endpoints', 'grain_junctions', 'total_branch_lengths', 'num_crossings', 'avg_crossing_confidence', 'min_crossing_confidence', 'num_mols', "total_contour_length", and 'average_end_to_end_distance'. Following training, the model was used to classify a mixed population of relaxed and supercoiled molecules immobilised in Ni, consisting of 52 molecules in total. Of these, 29 were classified as relaxed and 23 were classified as supercoiled.

**Topological determination.** The 3D trace coordinates obtained during the ordering process were arranged as the index, N, x-coordinate, X, y-coordinate, Y, and pseudo height, Z values into an NXYZ array. If a second trace was present in the same object, e.g., in DNA catenanes, the following traces were re-indexed from 0. The NXYZ array was then input to the Topoly library[65] using a Jones polynomial to calculate the topological species. This code is available on the TopoStats GitHub (https://github.com/AFM-SPM/TopoStats.git) and as a package on PyPi (topostats 2.3.0).

**Data cleanup.** Data directly output from TopoStats was cleaned according to the 'topo_utils' script used for topological classification. The cleanup steps are as follows: (1) The contour length row was empty - indicating the tracing pipeline failed. (2) The trace object contained linear molecules - due to poor masking/skeletonisation breaking up the trace. (3) The molecule contained greater than two crossing segments at a single crossing - this removed clustered crossings, which are unable to be resolved. (4) The number of molecules identified by the trace was not one for knots and two for catenanes - as all samples contain closed loops, any unpaired branches from inaccurate segmentation of close DNA duplexes manifested as small linear molecules or unexpected catenanes in the results.

For topological classification and their confidence rankings, steps 1, 2 and 3 were used because if the sample was unknown, the number of expected molecules in step 4 would also be unknown. For the calculation of contour length, steps 1, 2, 3 and 4 were applied, in order to separate the smaller and larger catenane molecules to ensure accurate length measurements. For the calculation of crossing distributions, steps 1, 2 and 4 were used to ensure we had full molecular representations. For surface compaction measurements, no additional steps were required as only the object masks and not the tracing pipeline were used. For catenane conformation classification, steps 1, 2, and 3 were performed. For the replication intermediate samples, all the TopoStats output data were used to obtain the total contour lengths. By forcing breaks in the molecular skeletons at odd-branched crossings via the configuration file, molecules that contain only 3 traces (two replicated regions and one unreplicated region) were filtered out to identify the unreplicated branch (most dissimilar in length) for analysis.

**Calculating contour length averages.** After data cleanup, a sample mean and error for each samples' contour lengths were obtained. For catenated samples, molecules were further separated into single topologies and smaller and larger molecules for a catenane topology. To get an average percentage error across multiple samples, an average of Eq. 2 using the true contour lengths and predicted contour lengths were used.

$$percentage\ error = \frac{|predicted - true|}{true} \times 100 \qquad (2)$$

**Molecular simulations.** *Model of the DNA molecule.* The catenated DNA was modelled by two intertwined coarse-grained discretized circular chains of interconnected beads with the diameter of 1 sigma. Each coarse-grained bead represented one turn of DNA 2.5 nm thick and contained 10.5 base-pairs[4]. The larger ring was made of $N \equiv N_L = 120$ beads and the smaller ring consisted of $N_S = 38$ beads.

The beads interacted via non-bonded interactions and bonded interactions (Supplementary Fig. 27). The non-bonded pair interactions were modelled by the WCA excluded volume interaction modelled by a fully repulsive truncated and shifted Lennard-Jones potential in the form of $U_{ex}(r) = 4\varepsilon_0\{[\sigma/(r-r_0)]^{12} - [\sigma/(r-r_0)]^6 + c\}$ if $r \leq 2^{1/6}$ and $U_{ex}(r) = 0$ otherwise, where $c = \frac{1}{4}$ and $r_0 = 1\sigma$[103]. Furthermore, we considered in the model also the electrostatic repulsion modelled by Debye-Hückel potential in the form $U_{DH}(r) = l_B\varepsilon_0 q_1 q_2 exp(-\kappa r)/r$, with Bjerrum length $l_B = 0.73$ nm $= 0.29\ \sigma$ and the Debye-Hückel screening length set to $\kappa = 0.81$. The values of $U_{ex}(r) + U_{DH}(r)$ were pre-calculated and tabulated to speed up computations. The calculation of the interactions was omitted for the nearest neighbours along the chain. The bonded interactions represented covalent bonds between the beads. The first of the bonded interaction is the harmonic potential between the beads in the form of $U_s(r) = k_s(r-r_0)^2$, where $k_s$ is the force constant that was set to 50 $\varepsilon_0 = 50\ k_BT$ and $r_0$ is the equilibrium distance set to 1 $\sigma$. Furthermore, the bonded interactions were represented by the three-body angular potential and four-body torsional potential. The angular

potential was implemented in the form of a harmonic potential $U_b(\theta) = 0.5\,k_b(\theta\text{-}\theta_0)^2$, where $k_b$ is the force constant that corresponds to the energy penalty against angle bending, and $\theta_0$ is the equilibrium angle that was set to $\theta_0 = \pi$. The force constant of the bending potential was set to 15 $k_BT/\sigma$ so that the persistence length of the modelled DNA is 50 nm[104] and the value was also corrected for the influence of bending stiffness introduced by electrostatic interaction. In order to involve torsional stiffness, new virtual beads were added following the approach we introduced earlier[105–107]. The virtual beads did not interact via pair potentials, but they do exhibit mass and hydrodynamic drag. There were 5 virtual beads introduced for each real bead. First, a virtual bead was placed axially in between two real beads and bound by strong harmonic potentials. Furthermore, 4 virtual beads ($p_{1i}$, $p_{2i}$, $p_{3i}$, $p_{4i}$; i $\epsilon$ 1,...,$N_L \wedge N_S$) were placed periaxially around the first virtual bead, hence forming a cross with the arm length set to 0.5 $\sigma$. The arms of the cross were locked with the dihedral potential along the chain, where the harmonic dihedral potential was introduced in the form $U_D(\phi)=k_D(\phi\text{-}\phi_0)^2$, where $\varphi$ is the dihedral angle formed by seceding periaxial beads $\phi(a_i, p_{1i}, a_{i+1}, p_{1i+1})$ and $\phi_0$ is the equilibrium angle set to $\phi_0 = 0$. For our models, we reduce underwinding by altering the number of turns defined by the equilibrium dihedral angle, $\phi \leq 2\pi\Delta Lk/N$. The underlying physical mechanisms may include electrostatic screening of the phosphate group charges, mechanical distortion and flattening of DNA on the surface, or partial dehydration due to water layer depletion between the DNA and the substrate. In our model, we use a coarse-grained approach to capture conformational phenomena on larger length scales. However, this model does not allow exploration of atomistic structural changes, which are instead incorporated through parameter adjustments, such as the equilibrium angle, in a top-down manner. We found the setting of $\phi$ for $\Delta Lk$ being 90-100 percent of the initial value produces consistent images with the AFM images and analyses, where the simulated supercoiled molecules exhibit much more open conformations (Fig. 6b). The force constant $k_D$ represents penalty against torsional deformation/twisting of the chain and it was set to $40\varepsilon_0$, so that the ratio between resulting writhe and twist Wr:Tw is about 66% in favour of the writhe.

**Molecular dynamics simulations.** We carried out Langevin molecular dynamics simulations by solving Langevin equations of motion for each bead $m\ddot{r} = -\nabla U(r) - \gamma m\dot{r} + R(t)(2\varepsilon_0 m\gamma)^{0.5}$, where each term in the equation represents the force acting on the bead, $-\nabla U(\mathbf{r})$ is given by the molecular potential interactions, $-\gamma m\dot{r}$ represents the friction, and $R(t)(2\varepsilon_0 m\gamma)^{0.5}$ is the random kicking force, where R(t) is a delta-correlated stationary Gaussian process. After the initial structures were generated, we performed long simulation runs of $2\times10^8$ simulation steps with the size of the integration step $\Delta\tau = 0.002$ time units until the gyration radius of the larger of the rings was equilibrated. The physical units of time are given as $\tau = \sigma\,(m/\varepsilon_0)^{0.5}$. We performed 100 independent simulations for each setting. The simulations were performed by using Extensible Simulation Package for Soft Matter[108,109].

**Modelling of DNA topology.** The topology of the DNA chain was modelled as a $4^2_1$ catenane - a 4-node catenane in which the two circles are intertwined in a right-handed path around each other. The initial conformation of the catenane was generated by using a set of parametric equations. In the initial conformation, the centre of masses of the smaller and the larger ring overlap. After the generation of the real beads, the virtual beads were added by using the previously reported algorithm. During this phase, the crosses formed by the periaxial beads were rotated so that the desired excess of the linking number $\Delta Lk$ in terms of the twist $\Delta Tw$ was imposed. In the case of simulations of the nicked DNA molecules, the dihedral lock along the molecule was interrupted at the chosen position by setting the penalty against torsional deformation/twisting $k_D = 0$. This resulted in the generation of a

positive supercoil on the larger ring during the equilibration period, which was mostly preserved during the deposition. In the case of the supercoiled molecules, the $\Delta Lk$ parameter for the larger ring was set to −4 and $\Delta Lk$ imposed to the smaller ring is -1. In order to account for the impact of the surface immobilisation effects observed in the AFM experiments, which exhibited a decreased level of observed supercoiling-induced writhe, the level of writhe was adjusted by re-establishing the equilibrium dihedral angle in the dihedral lock (Fig. 6a–c). The maximum value of the new equilibrium angle considered was limited to $\varphi_0^* \leq 2\pi\,\Delta Lk/N$. The simulations where the change of the equilibrium angle compensates for the initial $\Delta Lk$ were compared with the nicked ones.

**Soft deposition onto the surface.** After the equilibration period, the shortest distance between any of the coarse-grained beads and the surface of the wall was calculated as $d = \min|r_i; z = 0|$ for i = 1,...,$N_L \wedge N_S$. When the shortest distance was found, all the coordinates of the discretized chains were treated by extracting the calculated distance $d$, effectively placing the catenane to the very vicinity of the surface. Next, another long simulation of $2\times10^8$ integration steps took place while the beads were held only by the short ranged attractive force of the surface modelled by Debye-Hückel potential under stipulation that each of the DNA beads having charge −2e is attracted by a divalent counterion with charge +2e in environment with Bjerrum length $l_B = 0.28$ and the Debye-Hückel screening length set to $\kappa = 0.8$[110]. In the case of supercoiled molecules with the adjusted equilibrium angle $\varphi_0^* \leq 2\pi\,\Delta Lk/N$, additional $2\times10^7$ integration steps were performed at the beginning of the soft deposition period.

**Statistical analysis of simulations.** Several metrics were calculated from the simulations to quantify the geometric and topological properties of both the large and small circles within supercoiled and nicked catenanes, namely twist, writhe, linking number difference ($\Delta Lk = Lk - Lk0$), radius of gyration and the distance between their centres of mass. To calculate the twist and writhe, we have used a routine developed earlier (available on GitHub at www.github.com/fbenedett/polymer-libraries). The routine is based on the discrete approximation of the Gauss linking integral[37], which evaluates pairwise interactions between segments of the curve. For twist, we use the angular deviation of local frames constructed along the curve. The resulting value of the twist takes into account the transformation of reinstalling the equilibrium dihedral angle described above. Writhe is calculated using the Gauss linking integral, where positive and negative values represent right-handed and left-handed crossings, respectively. Twist is determined by integrating the angular rotation of the polymer's local reference frame along its length, with positive and negative values reflecting right-handed and left-handed twists. The sign conventions for both quantities follow standard definitions in polymer physics and are based on the geometric orientation of the polymer chain in space.

**Principal Component Analysis (PCA).** To determine differences in properties of nicked and supercoiled molecules, as well as differences induced following molecule adsorption, Principal Component Analysis (PCA) was performed on just the final structures of each simulation trajectory to avoid unwanted correlation that could arise from including multiple frames from the same trajectory. Data were normalised prior to performing PCA to avoid larger-scale metrics from dominating the analysis. To further probe conformational differences between nicked and supercoiled catenanes adsorbed on the surface, the number of self-crossings and catenated crossings was computed geometrically by projecting line segments onto an XY plane. Here, self-crossings refer to instances when either the large or small catenated circle intersects with itself, whilst catenated crossings refer to inter-

molecule intersections. The frequency of each crossing type was calculated, as well as the Euclidean distance between crossings of the same type.

**Pseudo-AFM generation.** Coarse-grained simulations were converted into "pseudo-AFM" images through a series of processing steps designed to replicate tip convolution and colour maps associated with experimental data, making visual comparison between simulations and AFM images easier. Coarse-grained simulations were converted into 2D heightmaps by projection onto the XY plane, with each coordinate coloured by height in nanometres. These heightmaps were then dilated to simulate the 5 nm tip radius associated with the AFM images presented throughout, allowing for visualisation of clustered crossings that were observed within the experimental data. Finally, Gaussian filtering was applied to smooth the structures, mimicking the resolution typically observed in AFM images. Qualitative comparisons between pseudo-AFM and experimental AFM images were used to guide simulation parameters based on their visual similarity, with greater similarity indicating optimised parameters.

### Reporting summary
Further information on research design is available in the Nature Portfolio Reporting Summary linked to this article.

## Data availability
All data used in this publication, and plotting scripts are publicly available on Figshare with https://doi.org/10.15131/shef.data.27143238[111]. Source data are provided with this paper.

## Code availability
All code developed in this publication is publicly available on Github via TopoStats v2.3.0 https://github.com/AFM-SPM/TopoStats, https://doi.org/10.15131/shef.data.22633528.v2[112].

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

## Acknowledgements

This work was supported by the Leverhulme Trust Research Programme Grant RP2013K-017 to S.D.C., a UKRI Future Leaders Fellowship MR/W00738X/1 to A.L.B.P, the Grant Agency of the Ministry of Education, Science, Research and Sport of the Slovak Republic (VEGA 2/0038/24 - Polymers with Active Chiral Topology and Nanotechnology) to R.R. and D.R, an UK Microbiology Society Research Visit Grant GA000994 to J.I.P., an ERC Marie Sklodowska-Curie Fellowship (SinMolTermination - 794962) to N.S.G. and a Wellcome Trust Investigator in Science Award (215510/Z/19/Z) to A.G. We wish to acknowledge the Henry Royce Institute for Advanced Materials, funded through EPSRC grants EP/R00661X/1, EP/S019367/1, EP/P02470X/1 and EP/P025285/1 and Robert Moorehead and Xinyue Chen for Dimension FastScan access and sup-port at Royce@Sheffield; computational resources from the National competence centre for high performance computing funded by the European Regional Development Fund (11070AKF2) to R.R. and D.R. We thank Dorothy Buck and Andrzej Stasiak for scientific discussion during the project and Tony Maxwell and Jamie Hobbs for reading and com-menting on the final version of the manuscript. We thank Aggeliki Skagia

for help with the purification of proteins for *Xenopus* experiments and Kenneth Marians and James Dewar for the provision of plasmids and experimental protocols.

## Author contributions

Formal contributions in authorship order (CrediT taxonomy): Conceptualisation: E.P.H., M.C.G., J.I.P., N.S.G., A.G., S.D.C., and A.L.B.P.; Data curation: M.C.G., L.W., S.W., N.S., N.S.G. Formal Analysis: E.P.H., M.C.G., L.W., S.W., T.E.C., R.R., K.H.S.M., N.S., H.E.B., N.S.G., D.R., and A.L.B.P.; Funding acquisition: S.D.C., N.S.G., A.G. and A.L.B.P.; Investigation: E.P.H., J.I.P., R.R., T.E.C., K.H.S.M., D.R. and A.L.B.P.; Methodology: E.P.H., M.C.G., J.I.P., L.W., S.W., T.E.C., N.S., H.B., N.S.G. S.D.C. and A.L.B.P.; Project administration: A.L.B.P; Resources: J.I.P., D.R., N.S.G., A.G., S.D.C., and A.L.B.P.; Software: M.C.G., L.W., S.W. and N.S.; Supervision: H.E.B., N.S.G., A.G., S.D.C., and A.L.B.P.; Visualisation: E.P.H., M.C.G., J.I.P., L.W., T.E.C.; Writing – original draft: E.P.H., M.C.G., J.I.P., S.D.C. and A.L.B.P.; Writing – review & editing: E.P.H., M.C.G., J.I.P., L.W., R.R., S.W., T.E.C., N.S., H.E.B., N.S.G., A.G., D.R., S.D.C., and A.L.B.P.

## Competing interests

The authors declare no competing interests.
