## [Transparent Peer Review file · Nature Communications]

Quantifying Complexity in DNA structures with High Resolution Atomic Force Microscopy

Corresponding Author: Professor Alice Pyne

Version 0:

Reviewer comments:

Reviewer #1

(Remarks to the Author)

Review for the revised version of the manuscript "Under or Over? Tracing Complex DNA Structures with High Resolution Atomic Force Microscopy" by Holmes et al.

Generally, I dislike that the authors do not reproduce my first-round comments in their entirety in the point-by-point response. To make my work as a reviewer easier, I would like to be reminded without any selection of the authors what I commented about the prior manuscript version (thus I had to go back to my own record).

One of the comments that the authors have not responded to was:

"In summary, I recommend the manuscript for a methods focused journal, revising it to highlight the strengths of an unsupervised methodology. Limitations, such as the need to analyze isolated DNA molecules should be discussed, as several DNA strands from different molecules overlapping might be wrongly interpreted by an automated algorithm as different topological features." (disregard the part "In summary, I recommend the manuscript for a methods focused journal,")

I think this comment remains unaddressed. How does the algorithm distinguish between overlapping strands as the result of the topology of a single molecule, from overlapping strands of two molecules that have adsorbed on the mica overlapping each other? I assume this is not possible, and only isolated molecules can be analyzed. Please respond and discuss limitations.

Otherwise, the authors responded to my prior comments.

Additional question in revision:

The images in Figure 1e i to iv are all displayed in the same height range. However, while images in iii and iv display clear height signatures at the crossings, image ii does not. Why? Similarly weird differences in topography of crossing are in Figure 4d, crossings 3 and 5 have large topographies, while crossings 1 and 2 have essentially none. Figure 5a, crossings 1, 2, and 4 give large topographies, crossing 3 does not - my own brief topography analysis of the vertical strand in this crossing does essentially not show any elevation change as compared to before and after crossing. What is the explanation for this, and what does the algorithm do with these cases?

(Remarks on code availability)

Reviewer #2

(Remarks to the Author)

The authors have significantly improved their manuscript by providing clearer experimental and simulation details and reorganizing figure order, resulting in a more coherent flow. The revised manuscript is enjoyable to read, and I recommend it

for publication.

Two minor points could further enhance the manuscript:

1. Supplementary Fig. 19 contains a minor labeling error in the resolution notation: "a_{ii}-0.256, a_{iii}-0.512, a_{iv}-1.024, a_v-2.056, a_{vi}-4.128 nm/px". It appears that panel "a_{ii}" actually corresponds to the lowest resolution (4.128 nm/px), which should be corrected.
2. The fifth panel in Fig. 1d is somewhat confusing due to the colored contour and cross-section profiles. Specifically, the structure appears to have a self-crossing region, yet the cross-sectional profiles do not clearly highlight differences in the FWHM. Could the authors clarify how the determination of overpassing versus underpassing regions was made?

(Remarks on code availability)

The version of TopoStats is provided. I have no problem accessing the automatic pipeline as the manuscript used here.

Reviewer #3

(Remarks to the Author)

The Authors have done a reasonable job of addressing my concerns with the original submission. Overall the manuscript is improved, but it is still somewhat sprawling and hard to follow. This is in part due to the impressive amount of data that spans multiple systems and the inclusion of simulations. Some of the descriptions of methods and processes are terse to the point of being difficult to parse. For example, the calculation of the knot classification probability remains opaque. I would encourage the authors to improve the clarity and logical flow of the manuscript, particularly since the new software is impressive and possibly transformative and the results open up new approaches to measure the complex topology of DNA knots and catenanes.

I have some specific points to consider below:

P2 3rd paragraph "for" should be deleted.

Fig s 9 – what are PC1 and PC2? Of the many variables considered, which correspond to PC1 and PC2?

P9 and Supplementary Fig 18 – the method used to estimate the probability of correctly classifying different knot structures should be more fully explained – the combinatorics in Supplementary fig 18 are not described with sufficient detail to shed light on the process. This is an important aspect of the work. I appreciate that the authors have thought carefully about how best to calculate the probability of correctly assigning knotted structures, but they should provide more details on the process. In Supp Fig. 18 the sign changes and counts provided in the table are not well defined – nor is how they connect back to the calculated probability.

P10. The discussion of the simulations of the attractive force between the DNA and the surface are somewhat confusing. The language, such as "increasing the adsorptive force" gives the impression that the force is tuned or varied in the simulations, when in fact it seems that the adsorptive force is effectively toggled rather than varied in a continuous manner. Clarifying this point in the section on page 10 would help resolve some confusion for the reader.

P 13. The classification of nicked and supercoiled DNA species should include a reference to figure S9. In figure S9 the classification ratio should be indicated and some reference should be made to the methods where the classification approach is described.

There may be an issue with versions since there are differences between the changes reported in the rebuttal letter and the revised version of the manuscript.

This is valuable and important work - It should be published but it would be better received and more impactful if it were easier to read and some of the methods and approaches were spelled out in more detail.

(Remarks on code availability)

Reviewer #1 (Remarks to the Author):

Review for the revised version of the manuscript "Under or Over? Tracing Complex DNA Structures with High Resolution Atomic Force Microscopy" by Holmes et al.

One of the comments that the authors have not responded to was:

"In summary, I recommend the manuscript for a methods focused journal, revising it to highlight the strengths of an unsupervised methodology. Limitations, such as the need to analyze isolated DNA molecules should be discussed, as several DNA strands from different molecules overlapping might be wrongly interpreted by an automated algorithm as different topological features." (disregard the part "In summary, I recommend the manuscript for a methods focused journal,"). I think this comment remains unaddressed. How does the algorithm distinguish between overlapping strands as the result of the topology of a single molecule, from overlapping strands of two molecules that have adsorbed on the mica overlapping each other? I assume this is not possible, and only isolated molecules can be analyzed. Please respond and discuss limitations.

Though it is theoretically possible to segment and separate molecules which have adsorbed on top of one another, we did not observe these due to the concentration at which this study was performed. The ability to separate overlapping molecules will depend on the orientation of the molecules at the point where they overlap. If overlapping segments are nominally perpendicular to one another, these can be paired effectively and the individual molecules separated.

Otherwise, the authors responded to my prior comments.

Additional question in revision:

The images in Figure 1e i to iv are all displayed in the same height range. However, while images in iii and iv display clear height signatures at the crossings, image ii does not. Why? Similarly weird differences in topography of crossing are in Figure 4d, crossings 3 and 5 have large topographies, while crossings 1 and 2 have essentially none. Figure 5a, crossings 1, 2, and 4 give large topographies, crossing 3 does not - my own brief topography analysis of the vertical strand in this crossing does essentially not show any elevation change as compared to before and after crossing. What is the explanation for this, and what does the algorithm do with these cases?

This is indeed an interesting effect, and we observe large variations in the height of the overpassing DNA segment at crossings. We postulate that this may be due to "slotting" of the two helices together, but have no evidence for this and so have not elaborated further in the text.

It is for this reason that we opted to use the full width at half maximum (FWHM) to determine the crossing order, in place of area under the curve and hessian gradient methods. For example, in Figure 1eii we see both the overpassing strands are roughly the same height, but the strand labelled in green, remains more continuous in height over a longer period at the crossing. This allows the crossings to be assigned correctly using the FWHM. As the

height is much lower, the change in FWHM is not as pronounced and this is reflected in the average crossing confidence of 0.06 for the supercoiled molecule, 0.22 for the 5-node knot, and 0.29 for the 4-node catenane.

These data are available with the dataset, however we have not included them at this point in the paper, as it is prior to the introduction of the crossing confidence and crossing determination algorithms.

We have added the following to the discussion: “We can identify the over- and under-passing strand even in crossings that show minimal height variation using the full width at half maximum for each crossing segment and determine a pseudo-confidence to inform downstream classification.”

Reviewer #2 (Remarks to the Author):

The authors have significantly improved their manuscript by providing clearer experimental and simulation details and reorganizing figure order, resulting in a more coherent flow. The revised manuscript is enjoyable to read, and I recommend it for publication.

Two minor points could further enhance the manuscript:

1. Supplementary Fig. 19 contains a minor labeling error in the resolution notation: “a_{ii}-0.256, a_{iii}-0.512, a_{iv}-1.024, a_v-2.056, a_{vi}-4.128 nm/px”. It appears that panel “a_{ii}” actually corresponds to the lowest resolution (4.128 nm/px), which should be corrected.

We inverted the order of these in the paper. This mistake has now been corrected - we thank the reviewers for their observation.

2. The fifth panel in Fig. 1d is somewhat confusing due to the colored contour and cross-section profiles. Specifically, the structure appears to have a self-crossing region, yet the cross-sectional profiles do not clearly highlight differences in the FWHM. Could the authors clarify how the determination of overpassing versus underpassing regions was made?

This crossing is the only writhe or trivial crossing in this molecule, and so has been highlighted in blue - the colour of the larger molecule. We have edited the schematic to show the verpassing strand in dotted blue to match the hand-drawn traces.

As discussed above, we do see variations in the height of the over-passing segment, which can reduce the confidence of the crossing assignment. However as the crossing assignment is based on the full width at half maximum (FWHM), the crossing assignment is still valid even for molecules with similar heights. The over-passing strand (dotted blue), appears smoother for longer in the underlying AFM topography, which manifests as an increase in the FWHM. Though the average crossing confidence for the molecule is high at 0.46 however the confidence for the 5th crossing is lower at 0.28.

We have added the following to the discussion: “We can identify the over- and under-passing strand even in crossings that show minimal height variation using the full width at half maximum for each crossing segment and determine a pseudo-confidence to inform downstream classification.”

Reviewer #3 (Remarks to the Author):

The Authors have done a reasonable job of addressing my concerns with the original submission. Overall the manuscript is improved, but it is still somewhat sprawling and hard to follow. This is in part due to the impressive amount of data that spans multiple systems and the inclusion of simulations. Some of the descriptions of methods and processes are terse to the point of being difficult to parse. For example, the calculation of the knot classification probability remains opaque. I would encourage the authors to improve the clarity and logical flow of the manuscript, particularly since the new software is impressive and possibly transformative and the results open up new approaches to measure the complex topology of DNA knots and catenanes.

I have some specific points to consider below:

P2 3rd paragraph “for” should be deleted.

Removed

Fig s 9 – what are PC1 and PC2? Of the many variables considered, which correspond to PC1 and PC2?

We have improved Supplementary Figure 9 to include the PCA loadings for each PCA plot which show the relative feature importance, as “d, Feature contributions to PC1 and PC1 for each PCA plot as output from TopoStats. The largest contributions are from 'grain_junctions', 'aspect_ratio', 'radius_max', 'num_crossings', 'smallest_bounding_width', 'max_feret', 'total_contour_length' and 'total_branch_lengths'. Colour bar = -2 to 4 nm, scale bar = 50 nm.”

P9 and Supplementary Fig 18 – the method used to estimate the probability of correctly classifying different knot structures should be more fully explained – the combinatorics in Supplementary fig 18 are not described with sufficient detail to shed light on the process. This is an important aspect of the work. I appreciate that the authors have thought carefully about how best to calculate the probability of correctly assigning knotted structures, but they should provide more details on the process. In Supp Fig. 18 the sign changes and counts provided in the table are not well defined – nor is how they connect back to the calculated probability.

We have added additional methods for “Calculating Crossing Classification Probability” and “Topological determination” including a worked example for Supplementary Figure 18a to provide more context.

P10. The discussion of the simulations of the attractive force between the DNA and the surface are somewhat confusing. The language, such as “increasing the adsorptive force” gives the impression that the force is tuned or varied in the simulations, when in fact it seems that the adsorptive force is effectively toggled rather than varied in a continuous manner. Clarifying this point in the section on page 10 would help resolve some confusion for the reader.

We agree that this was phrased somewhat confusingly and have modified this section on page 10 to improve the clarity.

“The coarse-grained simulations indicate that the effect is dependent on the presence of an adsorptive force towards the surface, equivalent to an electrostatic potential. The adsorptive force increases the confinement strength and confinement free energy which results in increased colocalisation of the crossings. The adsorptive force also induces a change in the shape and appearance of the knots which tend to create rosette-like conformations (Supplementary Fig. 21e). Interestingly, the differences between twist and torus 5-node knots are less pronounced in the absence of the adsorptive force (Supplementary Fig. 21c,d).”

P 13. The classification of nicked and supercoiled DNA species should include a reference to figure S9. In figure S9 the classification ratio should be indicated and some reference should be made to the methods where the classification approach is described.

We have added a section to the methods on the Random Forest classifier that was used to computationally obtain the percentage of nicked vs supercoiled DNA in a population formed of 50% supercoiled molecules and 50% relaxed molecules. In brief, this classifier was trained on separate supercoiled and relaxed species immobilised using the same protocol, and then applied to the mixed population, obtaining a 56:44 classification.

We have also added a reference to Figure S9 to the discussion:

“We show that we can use these metrics to recover a 56:44 classification of a 50:50 mixed population of nicked and supercoiled DNA plasmids using random forest classification (Supplementary Fig. 9), highlighting the wide-scale applicability of our approach.”

There may be an issue with versions since there are differences between the changes reported in the rebuttal letter and the revised version of the manuscript.

We had added additional simulations of knotted DNA to Supplementary Figure 21 which weren't shown in the original submission, these simulations are referenced after Figure 4 in the text and provide validation that the conformation differences we observe for the catenated structures are also present in knotted structures and driven by the surface adsorption.

This is valuable and important work - It should be published but it would be better received and more impactful if it were easier to read and some of the methods and approaches were spelled out in more detail.

In addition to the methods referenced above, we have also added further information regarding data cleanup steps, including which steps were used in each section. In addition to the methods section, we have added an extensive documentation section to our codebase, designed to make this tool as usable as possible for the community <https://afm-spm.github.io/TopoStats/main/advanced.html>.